# Proteogenomic landscape of squamous cell lung cancer

Paul A. Stewart [1,2], Eric A. Welsh[2], Robbert J.C. Slebos [1], Bin Fang [3], Victoria Izumi[3], Matthew Chambers[2], Guolin Zhang[1], Ling Cen[2], Fredrik Pettersson[2], Yonghong Zhang[2], Zhihua Chen[2], Chia-Ho Cheng[2], Ram Thapa[2], Zachary Thompson[2], Katherine M. Fellows[1], Jewel M. Francis[1], James J. Saller[4], Tania Mesa[5], Chaomei Zhang[5], Sean Yoder[5], Gina M. DeNicola [6], Amer A. Beg[7], Theresa A. Boyle[4], Jamie K. Teer [8], Yian Ann Chen[8], John M. Koomen [9], Steven A. Eschrich [8] & Eric B. Haura[1]

How genomic and transcriptomic alterations affect the functional proteome in lung cancer is not fully understood. Here, we integrate DNA copy number, somatic mutations, RNA-sequencing, and expression proteomics in a cohort of 108 squamous cell lung cancer (SCC) patients. We identify three proteomic subtypes, two of which (Inflamed, Redox) comprise 87% of tumors. The Inflamed subtype is enriched with neutrophils, B-cells, and monocytes and expresses more *PD-1*. Redox tumours are enriched for oxidation-reduction and glutathione pathways and harbor more *NFE2L2/KEAP1* alterations and copy gain in the 3q2 locus. Proteomic subtypes are not associated with patient survival. However, B-cell-rich tertiary lymph node structures, more common in Inflamed, are associated with better survival. We identify metabolic vulnerabilities (*TP63*, *PSAT1*, and *TFRC*) in Redox. Our work provides a powerful resource for lung SCC biology and suggests therapeutic opportunities based on redox metabolism and immune cell infiltrates.

[1] Department of Thoracic Oncology, H. Lee Moffitt Cancer Center and Research Institute, Tampa, FL 33612, USA. [2] Biostatistics and Bioinformatics Shared Resource, H. Lee Moffitt Cancer Center and Research Institute, Tampa, FL 33612, USA. [3] Proteomics Core Facility, H. Lee Moffitt Cancer Center and Research Institute, Tampa, FL 33612, USA. [4] Department of Anatomical Pathology, H. Lee Moffitt Cancer Center and Research Institute, Tampa, FL 33612, USA. [5] Molecular Genomics Core Facility, H. Lee Moffitt Cancer Center and Research Institute, Tampa, FL 33612, USA. [6] Department of Cancer Physiology, H. Lee Moffitt Cancer Center and Research Institute, Tampa, FL 33612, USA. [7] Department of Immunology, H. Lee Moffitt Cancer Center and Research Institute, Tampa, FL 33612, USA. [8] Department of Biostatistics and Bioinformatics, H. Lee Moffitt Cancer Center and Research Institute, Tampa, FL 33612, USA. [9] Department of Molecular Oncology, H. Lee Moffitt Cancer Center and Research Institute, Tampa, FL 33612, USA. Correspondence and requests for materials should be addressed to E.B.H. (email: eric.haura@moffitt.org)

L ung cancer continues as the cause of the most cancer-related deaths in the United States and is a major health care concern throughout the world[1]. Recently, therapeutic options for the treatment of lung cancer have emerged through better understanding of the molecular mechanisms of tumor formation and progression. In adenocarcinoma of the lung (ADC), identification of somatic gene mutations, amplifications, or fusions of oncogenes, such as receptor tyrosine kinases, has facilitated development of targeted agents with small molecule kinase inhibitors or therapeutic monoclonal antibodies[2]. However, few inroads in targeted therapy have been made for squamous cell lung cancer (SCC), despite initial enthusiasm about targeting EGFR, FGFR, DDR2, and PI3K[2]. In contrast, immune checkpoint inhibitor therapy has demonstrated durable tumor regressions in SCC histologies with prolonged survival. This result has led to approval of multiple antibodies targeting the PD-1/PD-L1 interaction for patients with advanced lung cancer and now provides an alternative therapy beyond conventional cytotoxic chemotherapy for patients with advanced SCC[3].

Genomic and transcriptomic technologies have enabled important insights into the molecular underpinnings of SCC, leading to initial molecular classification strategies[4–6]. The Cancer Genome Atlas (TCGA) identified recurrent mutations in genes associated with cell cycle and apoptosis (TP53, CDKN2A, and RB1), antioxidant gene expression (NFE2L2, KEAP1), phosphatidylinositide 3-kinase signaling (PIK3CA, PTEN), and epigenetic signaling (MLL2)[4]. TCGA also identified high level changes in chromosome gain and loss associated with severe genomic instability. In addition to tumor autonomous features, patterns of infiltrating immune cell types have been associated with tumor progression and patient prognosis[7]. Based on these results, studies such as the NCI's Molecular Analysis for Therapy Choice (MATCH) trial are attempting to capitalize on improved molecular knowledge of SCC to employ precision medicine targeting PI3K, CDK4/6, FGFR, MET, and PD-L1.

These genomic and transcriptomic alterations shape the functional proteome, control infiltration of immune cells, and present potential vulnerabilities that can be therapeutically exploited, but only after their specific roles in these molecular mechanisms is known. To begin to address this lack of knowledge, we report an integrated analysis incorporating expression proteomics with DNA copy number variation (CNV), somatic mutations, and mRNA expression levels determined by RNAseq in 108 SCC tumors, which is further informed by accompanying patient outcomes, evaluation of tumor pathology, and other clinically relevant data. The incorporation of mass spectrometry-based proteomics data is a critical addition as protein abundance can correlate poorly with corresponding mRNA abundance[8–11]. Our study leverages prior deep genomic and transcriptomic studies of SCC allowing a focused examination of the SCC proteome and its relationship to previously observed genomic or transcriptomic subgroups[4–6]. Finally, we discuss how the knowledge gained from proteogenomics can create a molecular classification with the potential to impact treatment strategies for SCC patients.

## Results

**Clinical and molecular features**. We identified 108 SCC snap frozen tumor tissues from patients consented under the Total Cancer Care™ protocol. We linked these samples to all available molecular, clinical, pathology, and outcomes data (Table 1, Supplementary Data 1, Supplementary Fig. 1). Patient clinical data and physician notes were reviewed to ensure these samples constituted treatment-naïve, primary lung cancer. Clinical characteristics were typical of this tumor type with older patients

(mean age = 69.1) and a majority of the cohort having a history of tobacco smoking (97.2%). Tumors were collected from patients with surgically resected stage I–III SCC of the lung (AJCC Version 7). The median follow-up of this cohort for overall survival was 4.8 years, while recurrence-free survival had a median of 4.2 years. Snap frozen tumor tissues were matched to a corresponding formalin fixed paraffin embedded sample for both image capture for hematoxylin and eosin (H&E) stained slides and immunohistochemistry assays. Snap frozen tumor tissues were randomized, pulverized into homogenized powder while still frozen, and split into two equal aliquots for targeted exon sequencing, copy number analysis, RNAseq, and mass spectrometry-based proteomic analysis.

Targeted exome sequencing of 154 genes revealed mutation patterns similar to those found in the TCGA patient cohort (Fig. 1)[4]. Copy number analysis revealed 5819 amplifications in 719 regions and 3884 losses in 666 regions, consistent with the findings observed by TCGA (Supplementary Fig. 2)[4]. Proteomic analysis was performed by liquid chromatography-tandem mass spectrometry (LC-MS/MS) using tandem mass tag (TMT) chemical labeling experiments, which gives comparable biological content to label-free LC-MS/MS (Supplementary Fig. 3)[12]. Similar to Zhang et al. and Slebos et al., we employed a highly stringent, two-step filtering process and identified 8300 protein groups (average of 6570 protein groups per sample; see Methods)[10,13]. Functional analysis and downstream associations with DNA and mRNA were restricted to the 4880 protein groups that were observed in >90% of samples, resulting in a final protein false discovery rate of 1.3%. Supporting targeted exome sequencing and copy number data can be found in Supplementary Data 2,

| Table 1 Squamous cell lung cancer cohort | |
|---|---|
| | **n = 108** |
| Age at diagnosis | 69.1 (8.3) |
| Gender | |
| Female | 36 (33.3%) |
| Male | 72 (66.7%) |
| Ethnicity | |
| Non-hispanic | 106 (98.1%) |
| Hispanic | 1 (0.9%) |
| Unknown | 1 (0.9%) |
| Race | |
| White | 105 (97.2%) |
| Black | 3 (2.8%) |
| AJCC-7 Stage | |
| I | 49 (45.4%) |
| II | 46 (42.6%) |
| III | 13 (12.0%) |
| Grade/differentiation | |
| Poor | 54 (50.0%) |
| Moderate | 51 (47.2%) |
| Well | 2 (1.9%) |
| N/A | 1 (0.9%) |
| Smoking | |
| Never-smoker | 3 (2.8%) |
| Ever/missing | 105 (97.2%) |
| Lymph nodes | |
| Negative | 79 (73.1%) |
| Positive | 29 (26.9%) |
| | **n = 104** |
| TLN Score | |
| 0 | 41 (39.4%) |
| 1 | 31 (29.8%) |
| 2 | 31 (29.8%) |
| 3 | 1 (1.0%) |

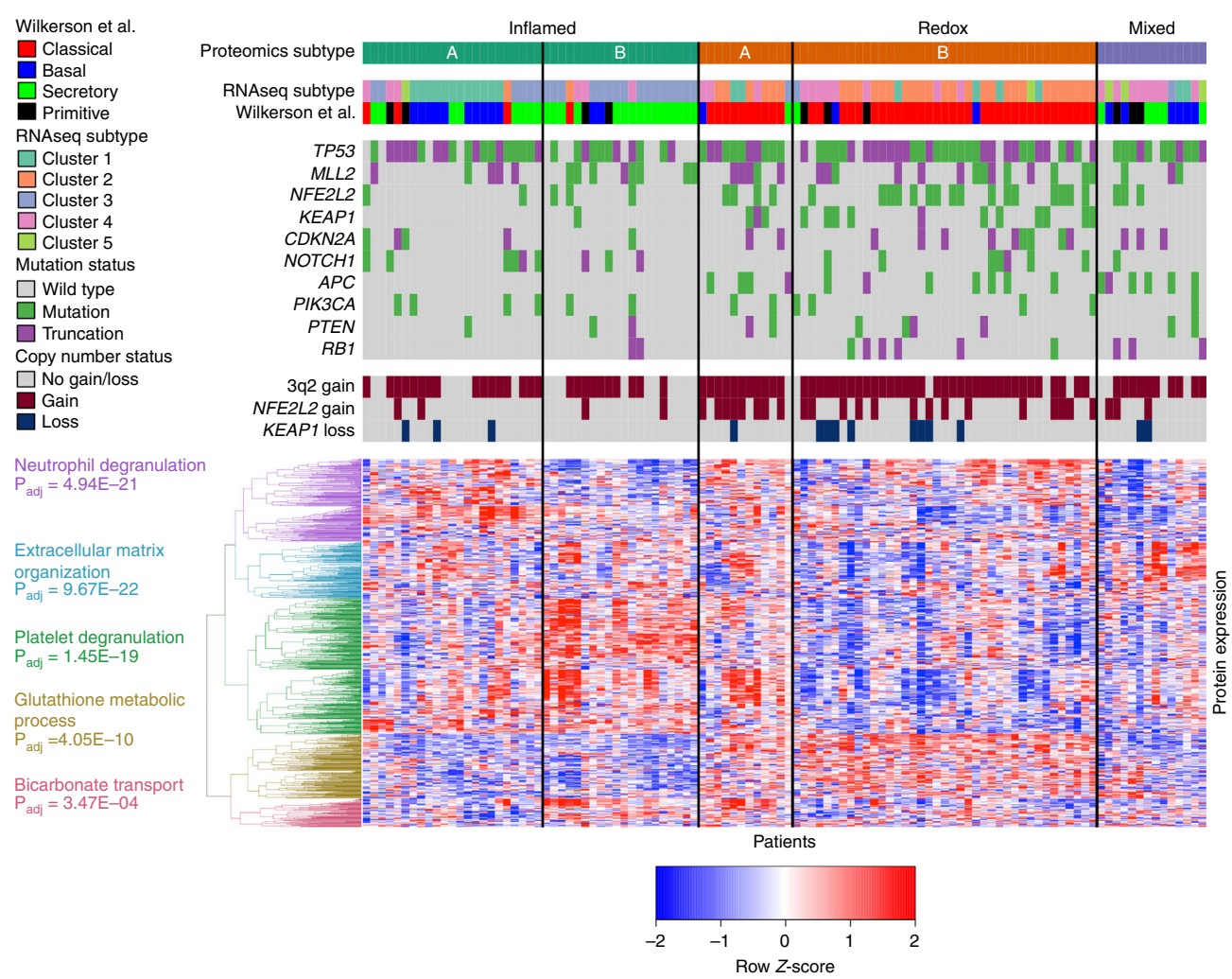

**Fig. 1** Identification of three proteomic subtypes of SCC. One hundred eight patient tumors are displayed as columns, and the 1000 most variable proteins by absolute median deviation are displayed as rows. The patient tumors were organized by consensus clustering into five clusters corresponding to three biological subtypes (Inflamed, Redox, and Mixed). There is partial concordance with the Wilkerson et al. mRNA-based classifiers of these same samples, but the primitive group is not recapitulated. Mutation status and copy alterations of commonly mutated SCC genes/loci are shown directly above the heatmap. We identified five groups of proteins from the heatmap clustering. We took the proteins from each group and searched against GO: Biological Processes to yield a list of pathways (Enrichr). The topmost enriched pathway with $P_{adj} \leq 0.05$ was used to label the protein clustering in the heatmap. The mean transcript-protein correlations for these pathways using matched RNAseq expression were: 0.46 for neutrophil degranulation, 0.56 for extracellular matrix organization, 0.35 for platelet degranulation, 0.64 for glutathione metabolic process, and 0.60 for bicarbonate transport

supporting proteomics data in Supplementary Data 3, and supporting RNAseq data in Supplementary Data 4.

**Inflammation and redox biology define tumor classification strategies using SCC proteomes.** Consensus clustering identified five proteomic clusters with clear patterns of expression (Fig. 1; Supplementary Fig. 4)[14]. Pathway analysis of the first two clusters revealed immune biology and were combined into an "Inflamed" subtype (43 tumors or 40%). The next two clusters showed clear oxidation-reduction (redox) biology and were combined into the "Redox" subtype (51 tumors or 47%). Notably, these first two subtypes make up 87% of the patient cohort. The fifth cluster includes biology associated with Wnt/stromal signaling and is referred to as the "Mixed" subtype (14 tumors or 13%). The subtypes were not associated with differences in clinical variables such as stage, gender, smoking status (Fisher Exact Tests $P$-values > 0.05; Supplementary Data 5), or tumor cellularity (Wilcoxon rank-sum $P$-values > 0.05; Supplementary Data 1). Each of these

subtypes will be detailed, but we first compared our identified subtypes to previously identified mRNA-based classifications[6].

**Relationship of the SCC proteome and transcriptome.** We first compare our proteomics subtypes to gene expression signatures of SCC previously identified by Wilkerson et al.[6]. RNAseq analysis was performed on the same homogenized tumor tissue samples as the proteomics, thus we can classify the Wilkerson et al. subtypes from gene expression profiling using RNAseq data. Within our cohort, 47 samples (44%) corresponded to the classical group, 30 (28%) to secretory, 21 (19%) to basal, and 10 (9%) to primitive, which is concordant with observations by both Wilkerson et al. and TCGA[4,6]. Our Inflamed proteomics subtype consisted of a mixture of mRNA groups but was predominantly secretory (immune) and basal (cell adhesion). The Redox proteomics subtype consisted primarily of the classical group (metabolism), and the Mixed proteomics subtype included a mixture of basal, secretory, and primitive tumors but none classified as classical. Interestingly, none of the three proteomic

subtypes were predominantly primitive. Independent consensus clustering of our RNAseq gene expression identified clusters similar to three of the four Wilkerson et al. mRNA groups: basal, secretory, and classical (Supplementary Fig. 5).

In single steady state measurements, mRNA levels have been shown to be poor predictors of protein abundance, and these findings could explain why our proteomic subtypes only partially recapitulate mRNA groups[8–11]. We assessed this concordance in our cohort by intersecting RNAseq and proteomics identifications, yielding 4625 matching transcript/protein pairs (Supplementary Fig. 6). Across the cohort, the 4625 matched transcript-protein pairs had an average Spearman's correlation ($\rho$) of 0.38, and this was comparable to previous observations in breast, colorectal, and ovarian cancers as well as in non-cancerous tissues[8–11]. To further investigate the underlying biology relating transcriptome and proteome, we compared the pathway enrichment of 1683 highly correlated transcript-protein pairs (Spearman's $\rho > 0.5$) to 1206 transcript-protein pairs that were poorly correlated (Spearman's $-0.2 < \rho < 0.2$). Highly correlated transcript-protein pairs were enriched ($P_{adj} < 0.05$) for neutrophil, extracellular matrix, apoptosis, oxidative stress, and glycolysis pathways. Poorly correlated pairs were enriched for nonsense-mediated decay and translation-related machinery, suggesting these functions are more regulated at the post-transcriptional and proteome levels (Supplementary Data 6).

**Neutrophil infiltration and activation define one Inflamed SCC cluster.** The first cluster (Fig. 1), Inflamed A, contained 23 tumors (21% of the cohort), and had significantly higher expression of neutrophil-associated proteins compared to the rest of the tumors (Wilcoxon rank-sum test, ±1.5 fold-change, $P_{adj} \leq 0.05$), including MPO, DEFA1, DEFA3, LTF, ELANE, MMP9, and RETN (Supplementary Data 7). We used a ±1.5-fold-change threshold for the expression proteomics data, because chemical labeling experiments can lead to underestimation of fold-changes[15]. S100 proteins related to extracellular matrix and neutrophil functions were also significantly higher. To better interpret these observations, we used a pathway enrichment approach combining Enrichr and MSigDB (Supplementary Data 8, Supplementary Data 9)[16,17]. Inflamed A was enriched for neutrophil degranulation (Enrichr $P_{adj} = 2.20E-24$) and matrisome pathways (MSigDB $P_{adj} = 5.06E-05$). Supporting protein differential expression data can be found in Supplementary Data 7.

**Antigen presentation superimposed on neutrophil biology defines the second Inflamed SCC cluster.** The second cluster, Inflamed B, contained 20 tumors (19% of tissues). Similar to Inflamed A, Inflamed B had significantly higher protein expression associated with neutrophil infiltrates and inflammatory response (AGER, SFTPA1, PIGR, and C3; Supplementary Data 7). To differentiate this cluster from Inflamed A, Inflamed B expressed significantly higher levels of 11 MHC Class II proteins and 9 cathepsins, which is consistent with antigen presentation biology. Supporting this observation were significantly elevated proteins related to γ-interferon and T-cell activation (CAV1, CD53, and IFI30). Pathway enrichment of Inflamed B indicated that neutrophil degranulation (Enrichr $P_{adj} = 1.52E-07$) and antigen processing pathways (MSigDB $P_{adj} = 8.42E-16$) were significant. Additionally, Inflamed B had enrichment of the same matrisome pathway (MSigDB $P_{adj} = 6.83E-12$) as Inflamed A.

**Inflamed subtype is enriched for infiltrating immune cells and has increased *PD*-1 gene expression.** Since the two Inflamed clusters shared similar immune and matrisome pathway enrichment, they were combined into a single Inflamed subtype for

subsequent analyses (Fig. 2a). Because our proteomic analysis indicated immune function, we used RNAseq data to infer infiltrating cell types. First, we applied Estimation of STromal and Immune cells in MAlignant Tumor tissues using Expression data (ESTIMATE) to infer tumor purity (Fig. 2b–d)[18]. Consistent with proteomic observations, the Inflamed subtype had the highest median Immune score. We next applied CIBERSORT to infer 22 human hematopoietic cell phenotypes (Fig. 2e–o)[7]. Inflamed had significantly higher proportions of memory B-cells (Wilcoxon $P = 5.73E-03$), monocytes (Wilcoxon $P = 3.78E-04$), and neutrophils (Wilcoxon $P = 0.021$) compared to the other subtypes, while plasma cells were the only cell type with significantly lower proportions in Inflamed (Wilcoxon $P = 0.048$). In a comparison of Inflamed A and Inflamed B, we found neutrophils significantly higher in Inflamed A (Wilcoxon $P = 0.025$) and regulatory T-cells significantly higher in Inflamed B (Wilcoxon $P = 0.039$). Resting NK cells and activated mast cells were higher in Inflamed A (Wilcoxon $P = 0.082$ and $P = 0.086$, respectively), and memory B-cells were higher in Inflamed B (Wilcoxon $P = 0.086$). Finally, to examine implications for immune checkpoint inhibitor therapy, we also examined expression of *PD-1* (*PDCD1*) and *PD-L1* (*CD274*) using the RNAseq data. Inflamed tumors have significantly higher *PD-1* mRNA expression (Wilcoxon $P = 1.74E-04$) than Redox and Mixed tumors, but *PD-L1* was not differentially expressed (Wilcoxon $P = 0.19$; Supplementary Fig. 7A–B). *PD-1* was higher in Inflamed B compared to Inflamed A (Wilcoxon $P = 0.099$), and there was no difference in *PD-L1* (Wilcoxon $P = 0.74$; Supplementary Fig. 7C-D). PD-1 and PD-L1 were not detected in the proteomics experiments. Taken together, these results indicate that Inflamed tumors have more contribution of immune cell types to the observed proteome expression. Supporting ESTIMATE and CIBERSORT data can be found in Supplementary Data 4.

We hypothesized that the neutrophil signatures generated by our proteomics, pathway, and CIBERSORT analyses could include not only mature neutrophils but also myeloid derived suppressor cells, monocytes, and macrophages. This point can have important ramifications for immunotherapies, since studies have observed that neutrophils dominate the immune landscape in lung cancer and have lymphocyte-suppressing capabilities[19]. To address this, we first confirmed our neutrophil finding with xCell, another method for estimating cell infiltration[20]. Similar to CIBERSORT, neutrophils were significantly higher in Inflamed (Wilcoxon $P = 2.083E-05$; Supplementary Fig. 7E) compared to the rest of the cohort. Next, we validated our finding by scoring both intratumoral and stromal CD33 using a dual color CD33 plus CD8 immunohistochemistry assay applied to 22 Inflamed A and 19 Inflamed B tumors. We observed good correlation between CD8 IHC staining and CD8 measured by mass spectrometry (Spearman's $\rho = 0.49$, $P = 3.60E-03$). We observed more CD33 positive cells in the tumor stroma (Fig. 3, Table 2) in the Inflamed A tumors while there were comparable amounts of intratumoral CD33 positive cells between Inflamed A and Inflamed B tumors. Overall, these results are consistent with Kargl et al. who found large areas of NSCLC infiltrated by CD45+ cells with nearly 50% of these cells being of myeloid lineage[19]. Together, these observations suggest that targeting CD33+ myeloid cells, using agents such as Gemtuzumab ozogamicin, could potentially augment immune checkpoint therapy by eliminating immunosuppressive cells from the tumor microenvironment.

**Inflamed tumors have fewer mutations of key genes and fewer chromosomal alterations.** We next sought to determine if any of the frequently observed gene mutations or copy number changes

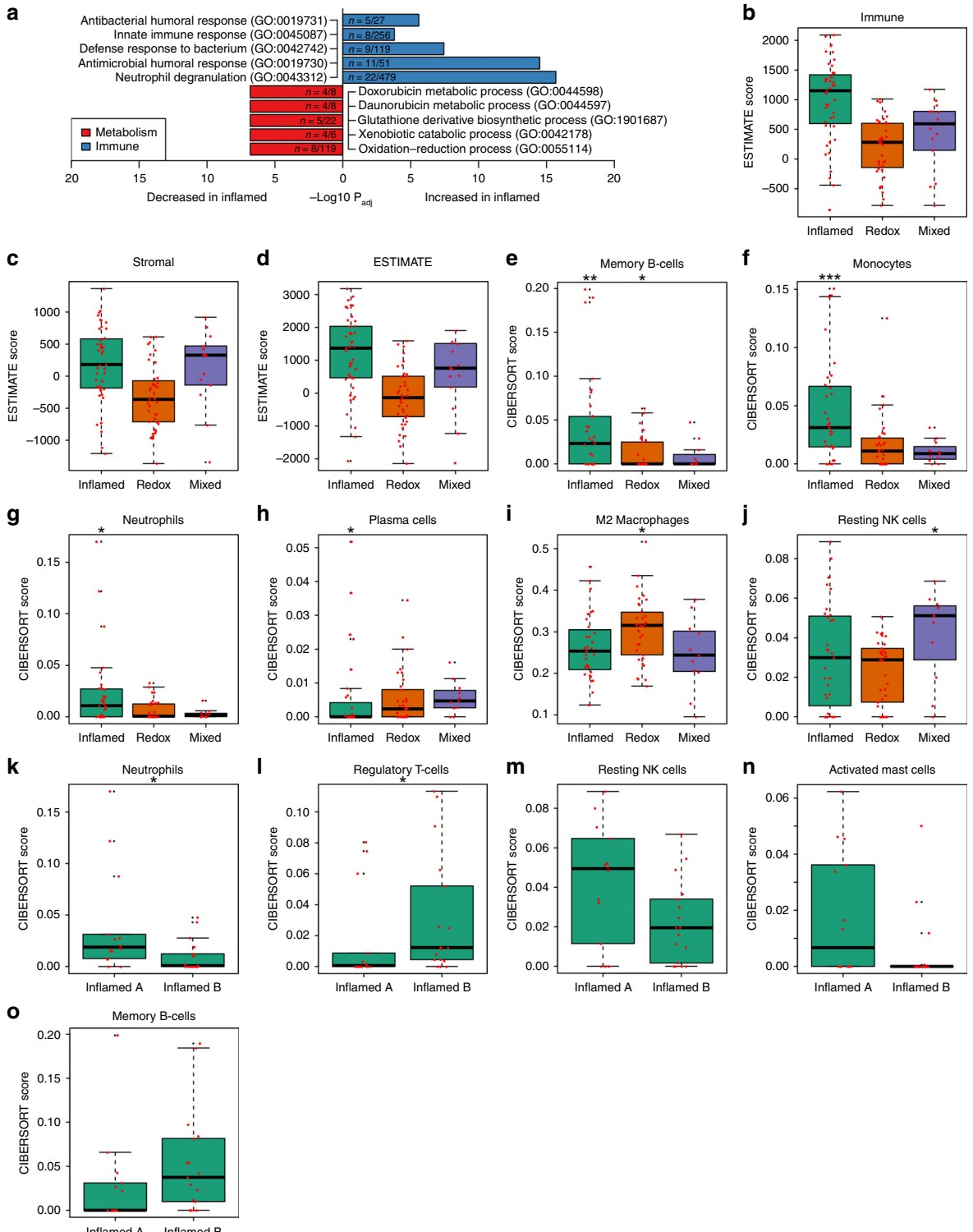

**Fig. 2** The Inflamed subtype is immune rich. **a** Pathway enrichment of significantly different proteins (±1.5 fold-change and Wilcoxon $P_{adj} \leq 0.05$ compared to the rest of the cohort). The Inflamed subtype was enriched for immune pathways including neutrophil degranulation. **b–d** Immune, Stromal, and ESTIMATE scores from the ESTIMATE algorithm[18]. The Inflamed subtype had the highest median Immune score. **e–o** Box plots showing CIBERSORT results for each immune subtype compared across the three proteomic subtypes and between Inflamed A and Inflamed B[7]. Inflamed had significantly higher proportions of memory B-cells (Wilcoxon $P = 5.73E-03$), monocytes (Wilcoxon $P = 3.78E-04$), and neutrophils (Wilcoxon $P = 0.021$) compared to the other subtypes, and plasma cells were significantly lower (Wilcoxon $P = 0.048$). M2 macrophages were significantly higher in Redox (Wilcoxon $P = 0.016$), and resting NK cells were higher in Mixed (Wilcoxon $P = 0.027$). Regulatory T-cells were significantly higher in Inflamed B (Wilcoxon $P = 0.039$), and neutrophils were significantly higher in Inflamed A (Wilcoxon $P = 0.025$). **box plots** Significance was denoted using the following: * = $P < 0.05$, ** = $P < 0.01$, *** = $P < 0.001$. The center line indicates the median, the bounds of the box indicate the interquartile range (IQR: defined as the difference between the 75th and 25th percentiles), the topmost and bottom-most horizontal lines indicate the most extreme points less than 1.5 times the IQR below the 25th or above the 75th percentile, black points indicate outliers, and red points indicate individual values

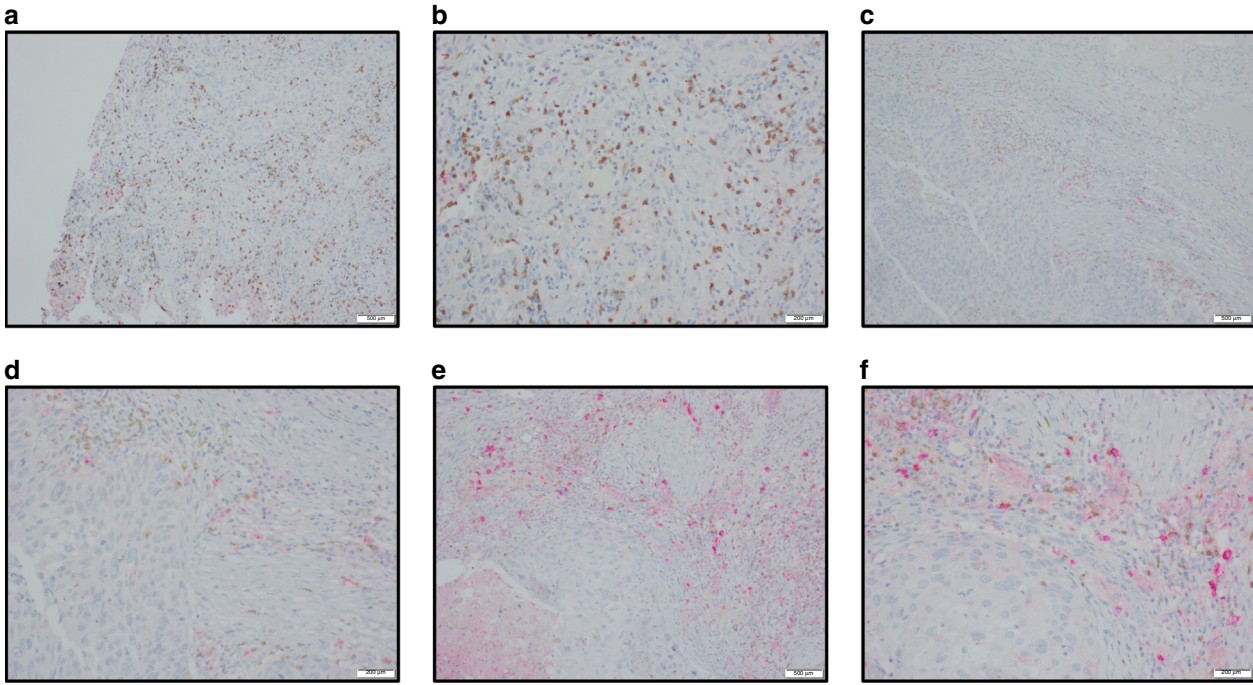

**Fig. 3** Scoring intratumoral and stromal myeloid lineage cells in Inflamed A and B. The IHC for case 1 at low (**a**) and high power magnification (**b**) highlights minimal CD33 + stromal neutrophils (in red) with a score of 1+ while background CD8+ stromal lymphocytes are highlighted in brown. The IHC for case 2 at low (**c**) and high power magnification (**d**) highlights a moderate CD33+ neutrophil population within the stroma with a score of 2 + with rare intratumoral CD33+ neutrophils. The IHC for case 3 at low (**e**) and high power magnification (**f**) illustrates the marked CD33+ neutrophil population within the stroma with a score of 3+. Scoring legend: 0 = virtual absence, 1 = low (<25%), 2 = moderate (25–50%), and 3 = marked increase (>50%)

**Table 2 CD33 immunohistochemistry scoring**

|  | Inflamed A (n = 21) | Inflamed B (n = 18) | P-value[a] |
|---|---|---|---|
| Stromal score |  |  | 2.91E-04 |
| 0 = virtual absence | 0 | 0 |  |
| 1 = low (<25%) | 0 | 4 |  |
| 2 = moderate (25–50%) | 0 | 6 |  |
| 3 = marked increase (>50%) | 21 | 8 |  |
| Intratumoral score |  |  | 0.12 |
| 0 = virtual absence | 0 | 2 |  |
| 1 = low (<25%) | 21 | 16 |  |
| 2 = moderate (25–50%) | 0 | 0 |  |
| 3 = marked increase (>50%) | 0 | 0 |  |

[a]Cochran-Mantel-Haenszel Test

contains *TP63/SOX2/PIK3CA*; OR = 0.28, Fisher's exact test P = 6.35E-03), 2q3 (containing *NFE2L2*; OR = 0.21, Fisher's exact test P = 2.22E-03), 12p1 (OR = 0.32, Fisher's exact test P = 5.83E-03), and 17q2 (OR = 0.41, Fisher's exact test P = 0.031). Inflamed had significantly fewer losses on 3p1 (OR = 0.29, Fisher's exact test P = 0.010), but did not have significantly more deletions of frequently altered regions when compared to the other subtypes. These results are consistent with reports of copy number variation negatively correlating with an immune score in a Korean SCC cohort[21]. Supporting mutation and copy number variation data can be found in Supplementary Data 2.

**Tertiary lymph node structures are associated with improved prognosis.** We next tested for associations of our three proteomic subtypes with the extensive clinical, pathology, and outcome data collected for our cohort, but there were no significant differences in overall or recurrence-free survival (Fig. 4a, b). This result was surprising given the immune phenotype observed in the Inflamed tumors, which we thought could be associated with better outcomes. We next tested top mutated genes (e.g., *TP53, MLL2,* and *NFE2L2*), but did not find any association with outcomes. We were also unable to identify associations with outcome in the DNA copy number, mRNA abundance, and protein abundance datasets. These negative findings are in line with the general lack of agreement of prognostic signatures in SCC[2]. We expect that as we learn how to target specific drivers of SCC biology that we will be able to define proteins and genes that can be used to predict patient outcomes.

We attempted to identify associations of patient outcome with DNA copy number, mRNA abundance, and protein abundance through a combined meta-analysis. A meta-analytic random-effects model was fitted using an empirical Bayes method for a heterogeneity estimator using gene-level CNV, RNAseq expression, and protein expression from 3484 genes present in all three

were associated with Inflamed tumors. The Inflamed subtype was less likely to have *NFE2L2* (OR = 0.24, Fisher's exact test P = 6.87E-03), *KEAP1* (OR = 0.09, Fisher's exact test P = 4.09E-03), or *APC* (OR = 0, Fisher's exact test P = 6.78E-05) mutations. The incidence of *TP53* mutations was reduced in Inflamed, approaching significance (OR = 0.42, Fisher's exact test P = 0.09). Data analyzed from TCGA showed a statistically significant reduction of *TP53* mutations in the Wilkerson secretory mRNA group (corresponding most closely to our Inflamed subtype) at similar proportions to the Inflamed subtype (OR = 0.38, Fisher's exact test P = 0.015)[4]. The Inflamed subtype was enriched for a single amplification on 14q3 (OR = 3.27, Fisher's exact test P = 0.012), but had significantly less amplification of 3q2 (which

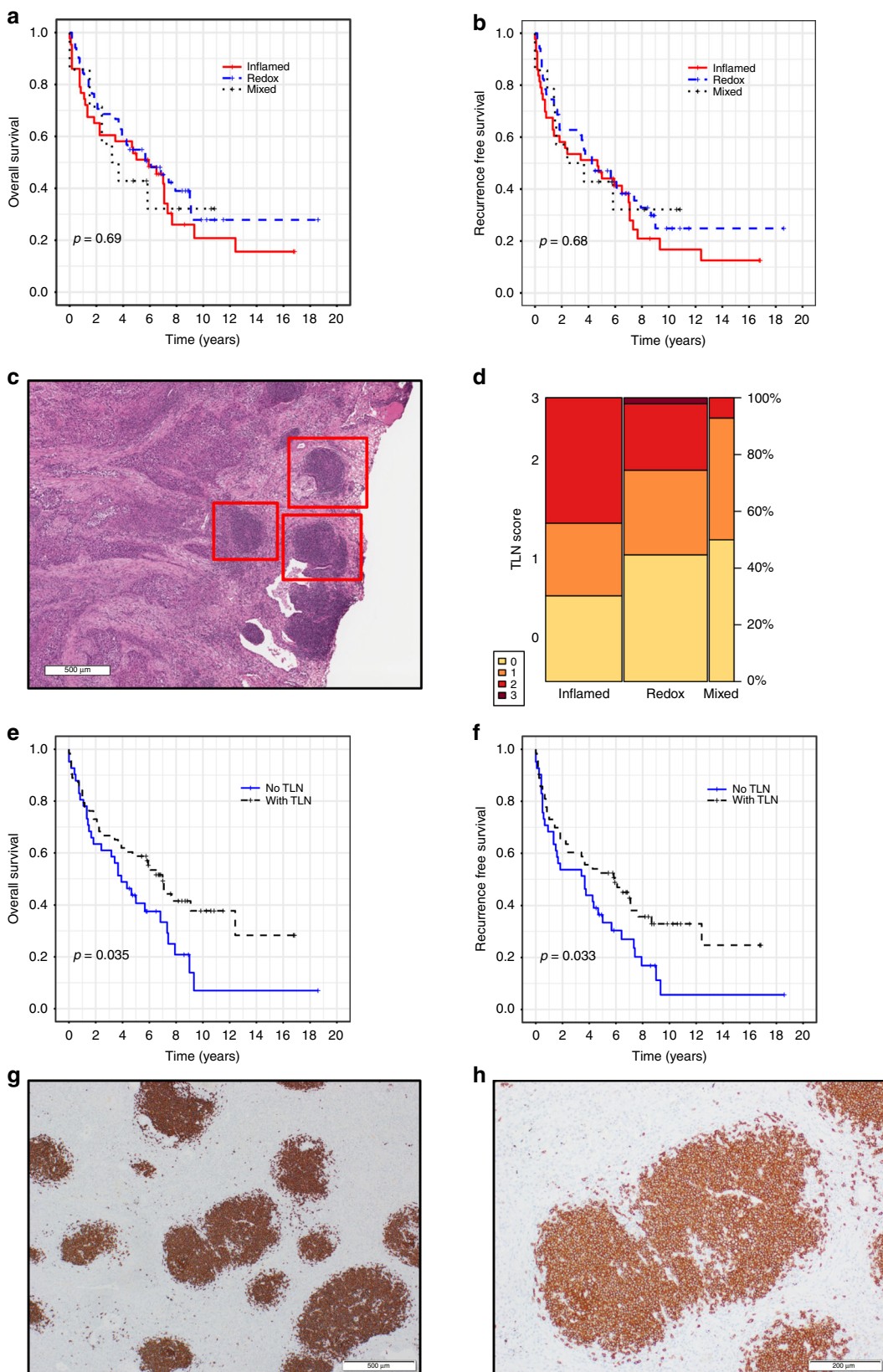

**Fig. 4** Tertiary lymph nodes (TLN) are enriched in the Inflamed subtype and are associated with better outcomes. **a, b** Proteomic subtypes are not associated with overall or recurrence-free survival. **c** TLN, indicated by red boxes, are observable in H&E stained slides under low magnification. **d** TLN can be observed in all three proteomic subtypes but are enriched in Inflamed (Cochran-Mantel-Haenszel P = 0.036). The height of each rectangle is proportional to the TLN score in a given subtype, and the width of each column is proportional to the number of samples in each subtype. **e, f** Presence of TLN provided a benefit to both overall and recurrence-free survival. **g, h** Immunohistochemical staining shows TLN primarily consist of CD20+ memory B-cells

datasets. However, we had to relax FDR to 30% (Storey's q-value < 0.3) in order to identify 15 proteins (Supplementary Fig. 8) associated with outcome. This meta-analysis associated high expression of the tyrosine phosphatase, PTPN12, with poor survival. PTPN12 has been described as a tumor suppressor that counteracts receptor tyrosine kinase signaling and oxidative stress[22], but it has also been shown to support tumorigenesis and was associated with poor prognosis in breast cancer[22]. We performed two exploratory analyses using PROGgene and cBioPortal to test if these 15 genes were associated with survival in the TCGA SCC cohort, but both failed to identify significant associations[23,24].

We hypothesized that the immune characteristics of the Inflamed subtype would also be observed in the histologic features of these tumors. Therefore, we first confirmed our memory B-cell finding using xCell[20]. Similar to CIBERSORT, memory B-cells were significantly higher in Inflamed (Wilcoxon P = 2.53E-04; Supplementary Fig. 7F) compared to the rest of the cohort. Next, we examined H&E slides and performed CD20 immunohistochemistry to detect B-cell infiltration from the same tumors used in our proteogenomic experiments. H&E slides (n = 104) revealed lymphoid cell aggregates consistent with tertiary lymph nodes (TLN), also known as tertiary lymphoid structures, near the cancerous tissues (Fig. 4c)[25]. We next scored H&E slides, blinded with respect to proteomic subtype, for TLN. TLN were significantly enriched in Inflamed (Cochran-Mantel-Haenszel P = 0.036, test; Fig. 4d), but TLN could be observed in all subtypes: 30 Inflamed tumors (69.7%), 26 Redox tumors (51%), and 7 Mixed tumors (50%). Importantly, there was significant association between the presence and absence of TLN (TLN score > 0 and TLN score = 0, respectively) in both recurrence-free survival (RFS; log-rank test P = 0.032) and overall survival (OS; log-rank test P = 0.035; Fig. 4e, f).

TLN exhibited strong staining for CD20 (n = 103), indicating the presence of high concentrations of B-cells (Fig. 4g, h). CD20 staining was significantly higher in tumors scored with TLN >= 2 compared to TLN = 0 (Wilcoxon P = 0.001). RNAseq showed significantly higher expression of B-cell markers, including BLK (log₂ ratio 1.31, Wilcoxon P = 1.45E-03), CD79A (log₂ ratio 1.55, Wilcoxon P = 1.37E-03), and CD79B (log₂ ratio 1.30, Wilcoxon P = 2.91E-03) in the TLN >= 2 group, and proteomics showed significantly higher expression of B-cell marker IGLL1 (0.63 log₂ ratio, Wilcoxon P = 0.013) in the TLN >= 2 group. CD20 staining was significantly higher in the Inflamed B subtype compared to Inflamed A (Wilcoxon P = 0.045), but no significant association was found between CD20 IHC scoring and either OS or RFS. CD20 staining was consistent with CIBERSORT findings that memory B-cells are significantly elevated in Inflamed (Fig. 2e). These results suggest that B-cell rich TLN, but not infiltrating B-cell numbers, are associated with better survival in surgically resected SCC. Taken together, these results suggest that patterns of immune cell infiltration and their organized structures are currently more effective for determining patient outcome than gene mutations, transcriptomic subclasses, or proteomic subclasses.

**The Redox proteome is driven by genomic alterations in the NFE2L2/KEAP1 complex.** The third cluster, Redox A, consisted of 12 samples (11%) and the fourth cluster, Redox B, consisted of 39 samples (36%). Both clusters shared elevated expression of aldo-keto reductase (AKR) and alcohol dehydrogenase family members. In a direct comparison, the only significant differences between Redox A and Redox B were an enrichment of blood-related pathways in Redox A. Since these differences between Redox A and Redox B were minor, they were combined into a single Redox subtype for subsequent analyses. Top elevated proteins of the combined Redox subtype included 5 AKR family members (Supplementary Data 7), which were also differentially expressed at the transcript level (Supplementary Data 4). AKRs are indicative of NFE2L2 (NRF2) activity and are overexpressed in both SCC and ADC[26]. The Redox subtype had significantly higher levels of TP63 expression and was enriched for oxidation-reduction, glutathione synthesis, and keratinization pathways (P_adj < 0.05; Fig. 5a).

We identified strong associations with copy number changes as well as NFE2L2 and KEAP1 mutations (Fig. 5b). Redox tumors had the most copy number gains (3680), followed by Inflamed (1402) and Mixed (737; Supplementary Fig. 2). Redox tumors also had the most copy number losses (2117) compared to Inflamed (1030) and Mixed (658). We tested the five regions with the most gains, the five regions with the most losses, and chromosome 2q3 (which contains NFE2L2). Redox was enriched for amplifications in 3q2 (contains TP63/SOX2/PIK3CA; OR = 4.32, Fisher's exact test P = 3.50E-03), 2q3 (contains NFE2L2; OR = 5.77, Fisher's exact test P = 2.00E-04), and 12p1 (OR = 2.61, Fisher's exact test P = 0.02). The Redox group was not enriched for losses compared to the other subtypes. The Redox subtype had increased numbers of KEAP1 (OR = 20.71, Fisher's exact test P = 1.12E-04) and NFE2L2 mutations (OR = 3.90, Fisher's exact test P = 3.99E-03). Twenty Redox samples (39% of Redox) possessed a NFE2L2 mutation and 14 samples (27% of Redox) possessed a KEAP1 mutation. Forty-three Redox samples (84%) had at least one type of genomic alteration of NFE2L2 or KEAP1 and 18 Redox samples (35%) had two or more. NFE2L2 amplifications were the most abundant (22 or 43%), followed by NFE2L2 mutations (20 or 39%), KEAP1 mutations (14 or 27%), and KEAP1 region loss (9 or 18%). To compare the mutant NFE2L2/KEAP1 proteome to the Redox subtype, we next took tumors with either a NFE2L2 or KEAP1 mutation (41 samples total; 32 Redox, six Inflamed, three Mixed) and compared to NFE2L2 and KEAP1 wild-type tumors. This stratification enriched for Redox tumors, and we observed downstream effects of NFE2L2 signaling, including oxidative stress and glutathione pathways (Fig. 5c).

KEAP1 is an E3 ligase that has multiple substrates, including NFE2L2. We hypothesized that tumors with KEAP1 mutations would be enriched for NFE2L2-independent co-expression, because the loss of KEAP1 ubiquitin ligase activity could result in accumulation of substrates. To test this hypothesis, we performed a protein-protein correlation analysis with KEAP1 protein expression in KEAP1 mutant tumors (15 samples total; 15 Redox, one Inflamed) and separately in KEAP1 wild-type tumors (93 samples total; 37 Redox, 42 Inflamed, 14 Mixed). There were no proteins significantly correlated with KEAP1 in the KEAP1 wild-type tumors nor in the KEAP1 mutant tumors (Spearman's ρ > 0.5 and P_adj ≤ 0.25). However, we identified 107 proteins (Supplementary Data 6) with a large difference in correlation to KEAP1 expression (Spearman's ρ > 0.5) between KEAP1 mutant and wild-type tumors, and pathway analysis identified regulation of mitotic cell cycle phase transition (Enrichr P_adj = 3.16E-04), which is consistent with KEAP1 function[27].

Given the importance of SOX2, TP63, and NFE2L2 transcriptional programs in SCC biology, we next examined mRNA transcript-protein correlations using MSigDB target signatures BENPORATH_SOX2_TARGETS, PEREZ_TP63_TARGETS, and NRF2_01, respectively (Table 3)[17]. All three transcription factor (TF) target signatures had positive transcript-protein correlations (average Spearman's ρ SOX2: 0.37, TP63: 0.34 and NFE2L2: 0.43), suggesting changes in TF target mRNA transcripts were similar to the overall transcript-protein correlations across the entire cohort (Spearman's ρ = 0.38). We observed a significant difference

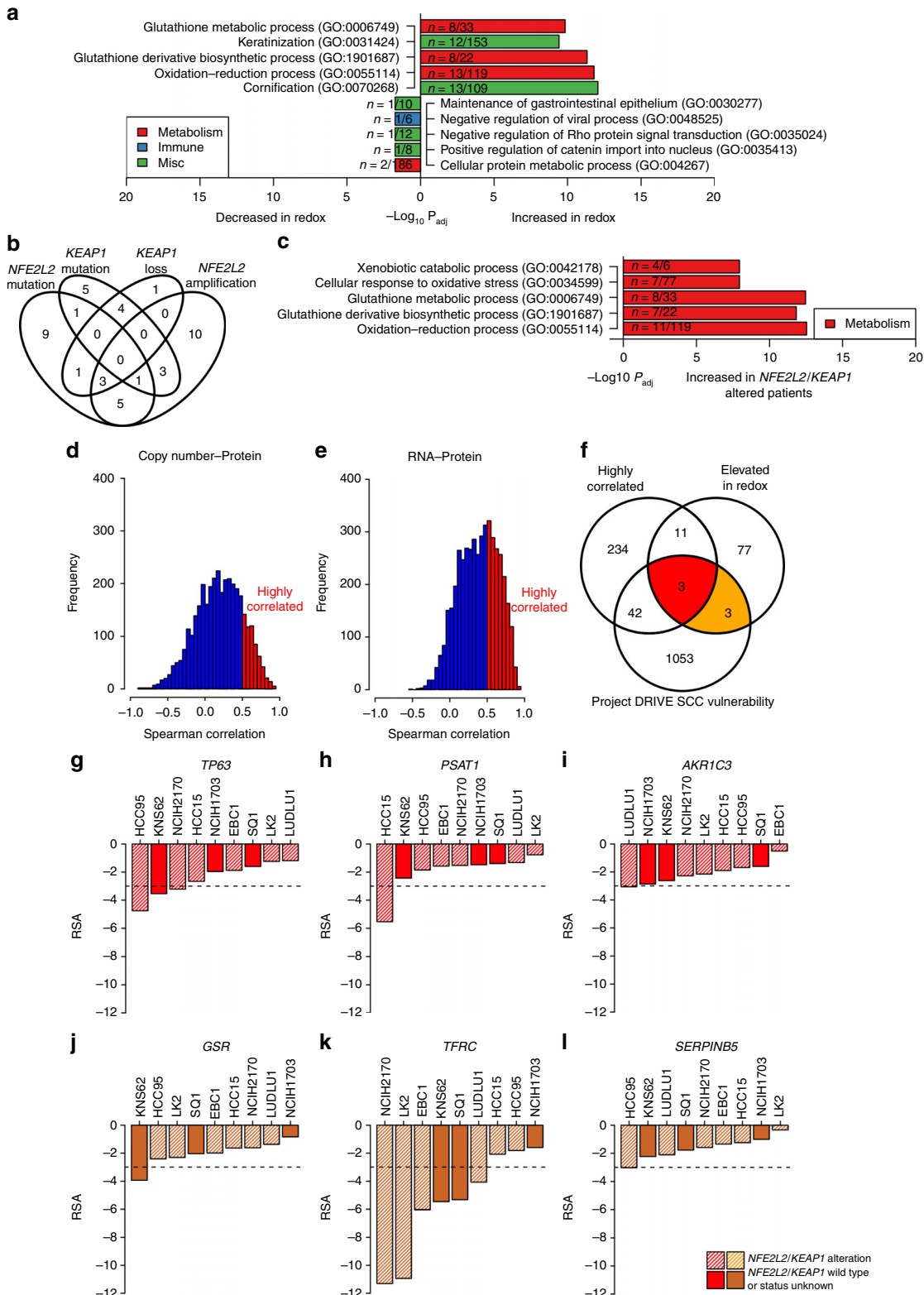

**Fig. 5** The Redox subtype is enriched for oxidative stress pathways and *NFE2L2/KEAP1* alterations. **a** Pathway enrichment of significantly different proteins (±1.5 fold-change and Wilcoxon P$_{adj}$ ≤ 0.05 compared to the rest of the cohort). The Redox subtype was enriched for oxidation-reduction and keratinization pathways. **b** 43 Redox samples (84%) had at least one genomic alteration of *NFE2L2* or *KEAP1*. **c** Stratifying patients by *NFE2L2* or *KEAP1* mutation (*NFE2L2/KEAP1* wild type patients vs. *NFE2L2* mutant and *KEAP1* mutant patients) revealed enrichment of oxidation-reduction and xenobiotic metabolism pathways in *NFE2L2/KEAP1*-altered patients (Enrichr). **d, e** We used DNA copy number to protein correlation >0.5 and correlation of RNA to protein >0.5 to identify 290 genes for subsequent analysis. **f** Highly correlated genes were intersected with significantly elevated proteins from the Redox subtype and genes that adversely impacted SCC viability in the Project DRIVE shRNA screen. **g–l** Project DRIVE screen results for squamous cell lung cancer cell lines. RSA < −3 is considered significant. Bar plots represent individual values

| Table 3 Average transcript-protein correlations of transcription factor targets | | | | | |
| --- | --- | --- | --- | --- | --- |
| Transcription factor | Overall | Inflamed | Redox | Mixed | P-value (Inflamed Vs Redox) |
| SOX2 | 0.37 | 0.31 | 0.37 | 0.39 | 3.59E−03 |
| NFE2L2 | 0.34 | 0.30 | 0.32 | 0.40 | 6.83E−01 |
| TP63 | 0.43 | 0.39 | 0.42 | 0.44 | 4.69E−01 |

between the SOX2 TF target correlations in Redox compared to Inflamed. No differences were observed in TP63 TF or NFE2L2 TF target correlations between these two groups. These results suggest that the SOX2 transcriptional program may be translated more consistently in the Redox subtype compared to the Inflamed subtype.

**TP63 and the NFE2L2-associated proteins PSAT1 and TFRC are vulnerabilities in SCC.** We hypothesized SCC therapeutic targets could be identified by integrating highly correlated DNA copy number/mRNA/protein changes with RNA interference screening data in SCC cell lines. Amplified genes that produce correspondingly higher amounts of proteins may be disease drivers and may represent vulnerabilities that can be exploited with therapeutic intervention. Because of the metabolism-related pathways found through integrative analysis and the identification of clear genomic drivers NFE2L2 and KEAP1, we focused our analysis on the Redox subtype. We identified 290 highly correlated proteins (Supplementary Data 6) by considering proteins with both copy number to protein correlation >0.5 and transcript to protein correlation >0.5 (Fig. 5d, e) and filtering by significance (Spearman's correlation $P_{adj} < 0.25$). We next identified 94 unique proteins significantly elevated in the Redox subtype. We then mined siRNA data from Project DRIVE and identified 1101 gene knockdowns that significantly impacted at least one SCC cell line (Supplementary Data 10)[28]. We used the same redundant siRNA activity (RSA) threshold of <−3 previously described as a cutoff for significance[28]. Finally, intersecting the highly correlated proteins, the proteins elevated in Redox, and the Project DRIVE SCC targets (Fig. 5f) identified AKR1C3, PSAT1, and TP63 in common (Fig. 5g–i). We also included three proteins elevated in Redox and also overlapped with Project DRIVE SCC genes for further investigation: GSR, TFRC, and SERPINB5 (Fig. 5j–l).

We performed an additional validation experiment by querying PICKLES for the same genes and cell lines used in our analysis of Project DRIVE results (five cell lines and five genes except SERPINB5 were available; Supplementary Fig. 9)[29]. We used a Bayes factor (BF) of >3 for a threshold of significance, as recommended[29]. The three genes consistently deleterious in both datasets, PSAT1, TP63, and TFRC, are biologically relevant to metabolic signaling within the Redox subtype and pose potential vulnerabilities in SCC. PSAT1 is regulated by NFE2L2 and catalyzes serine biosynthesis, which is important for the growth of NFE2L2/KEAP1-mutant lung cancer cell lines[30]. TP63 represents the ΔNp63 (or DNp63) isoform, which is the dominant isoform in SCC[4,31]. ΔNp63 has a missing N-terminus and altered exon use that produces an oncogenic protein[32]. TP63 amplification has been shown to upregulate glutathione metabolism, which is enriched in Redox (Fig. 5a), and promotes tumorigenesis[33]. Deletion of the ΔNp63 or ΔNp73 isoforms in TP53-deficient tumors led to metabolic reprogramming and tumor regression[34]. NFE2L2 regulates iron homeostasis, and in turn TFRC is an iron uptake receptor related to ferroptosis and may be a target in cancer[35]. Taken together, these results suggest that Redox tumors harbor metabolic vulnerabilities that could be therapeutically exploited.

**The Mixed subtype of SCC is associated with Wnt signaling and has increased stromal infiltration.** The fifth cluster, referred to as the Mixed subtype, was not as well defined through proteomic analysis as the Inflamed or Redox subtypes. The Mixed subtype exhibited significant decreases in oxidation-reduction and neutrophil pathways (Enrichr $P_{adj} < 0.05$; Fig. 6a) when compared to other tumors. Mixed was the smallest of the three subtypes (14 tumors or 13%), and only four proteins were significantly increased in this subtype: CHRAC1, FN1, MARCKSL1, and FHL2. Principal component analysis of protein expression data revealed the Mixed subtype was interspersed between the two clusters of Inflamed and Redox (Fig. 6b), which is consistent with the lack of pathway enrichment in this subtype. However, FN1, MARCKSL1, and FHL2 can be directly tied to Wnt/β-catenin signaling (CTNNB1)[36–38]. Consistent with our observations about protein abundance associated with Wnt signaling, the Mixed subtype had significantly more APC mutations (OR = 5.52, Fisher's exact test P = 8.78E-03; Fig. 6c) with six of the 14 tumors (43%) harboring alterations. One Mixed tumor with wild type APC harbored an oncogenic (S37F) CTNNB1 mutation[39], which was one of only two mutations in CTNNB1 observed in the cohort. We identified no significant enrichment for copy gains or losses associated with the Mixed subtype. Using ESTIMATE, Mixed tumors had the highest median Stromal score and the second highest median Immune and ESTIMATE scores (Fig. 2c, d). Tumor stroma have been shown to secrete Wnt ligands to activate Wnt signaling in cancer cells[40].

## Discussion

We report an examination of SCC centered on mass spectrometry-based proteomics integrated with parallel analyses of both DNA and mRNA to define molecular subtypes and identify key alterations that drive these tumors. To our knowledge, this study is the deepest analysis of genomic, transcriptomic, and proteomic datasets in lung cancer[41–43]. By identifying and quantifying large numbers of proteins in SCC tumors and incorporating bioinformatics approaches, we identified three subtypes of SCC at the proteomic level. Our results show SCC can be thought of as a disease with three subtypes, the bulk (87%) of which are associated with either immune infiltration (Inflamed) or oxidation-reduction (Redox) biology. This line of thinking is compelling, because it indicates that the majority of patients could benefit from therapies directed against immune cell types (Inflamed) or metabolic modulation of tumor intrinsic pathways (Redox). While the proteomic subtypes are not currently prognostic in surgically resected SCC, they may be more important in effecting patient outcomes in the context of immunotherapy, targeted therapy, and cytotoxic chemotherapy treatments.

One important observation here is the reduced immune infiltrates in Redox tumors, which suggests that NFE2L2/KEAP1 signaling may lead to tumor evasion of immune surveillance programs (Supplementary Fig. 10). One mechanism could be keratin-dependent features that mechanically interfere with immune cell infiltration. We found higher levels of keratin proteins associated with Redox tumors, which matches an observation made in KEAP1-deleted mouse models that suffer from hyperkeratosis of the esophagus and stomach[44]. High

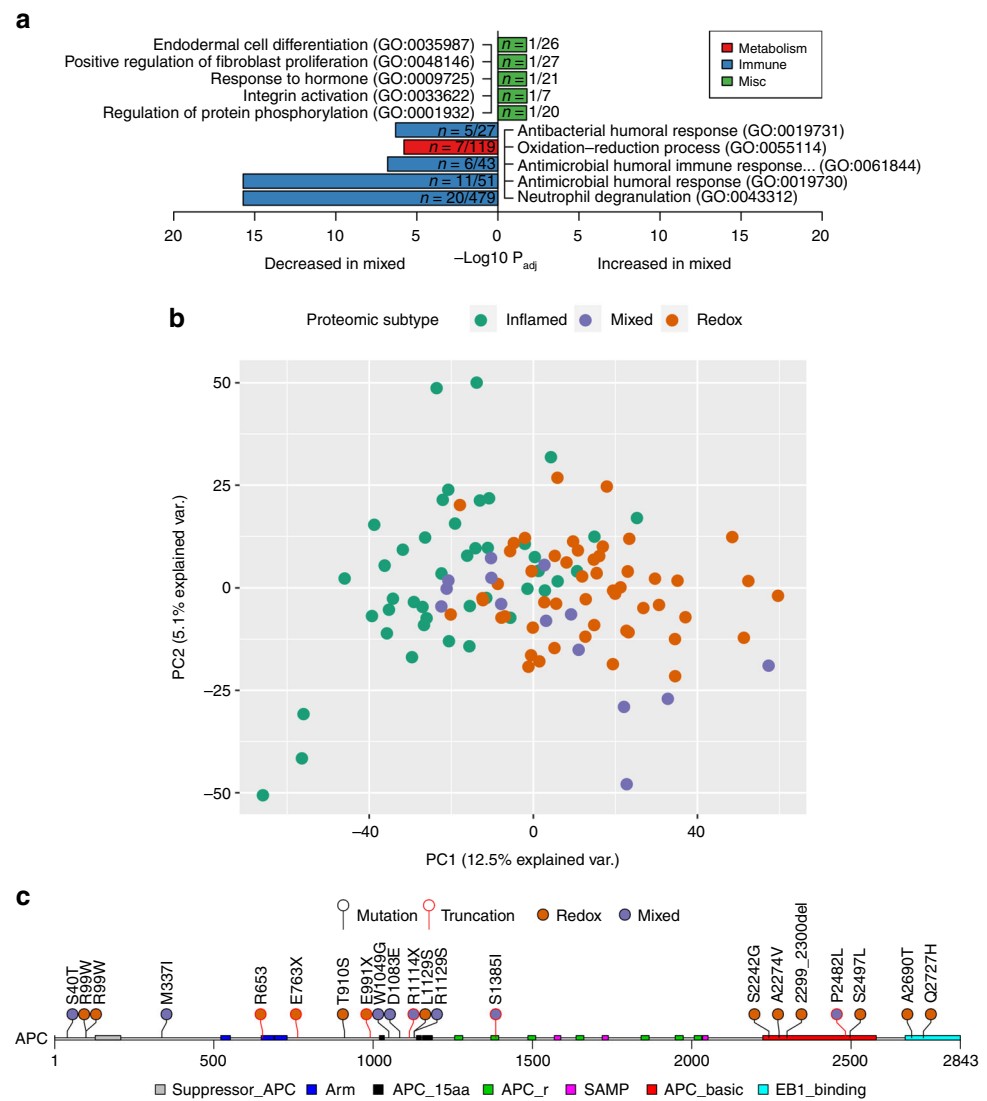

**Fig. 6** The Mixed subtype has biology associated with Wnt/stromal signaling. **a** Pathway enrichment of significantly different proteins (±1.5 fold-change and $P_{adj} < 0.05$ compared to the rest of the cohort). There were no significantly elevated pathways in Mixed, but several pathways were decreased including immune and oxidation-reduction pathways. **b** Principal component analysis of the three subtypes reveal distinct clustering of Inflamed to the left, Redox on the right, and Mixed interspersed throughout. **c** Lollipop plot of APC alterations in the cohort with PFAM domain annotation (https://pfam.xfam.org/) extracted from cBioPortal (http://www.cbioportal.org). Mutations with black lines were nonsynonymous and red lines were truncating. Inflamed had no *APC* mutations

matrix density, which can be affected by keratins, is associated with reduced migration of T cells[45]. Another potential mechanism could be related to directly to redox biology, since NFE2L2 is a key regulator of both the intracellular and extracellular redox states. The extracellular redox state is primarily regulated by the cystine/cysteine (CSSC/Cys) redox couple[46], which is reduced by the NFE2L2 target, SLC7A11 (xCT)[47]. The oxidized CSSC/Cys couple promotes oxidation of plasma membrane thiols, elevates peroxide levels, promotes cytokine secretion from monocytes[48], and promotes fibroblast proliferation and extracellular matrix production[49]. Further, peroxide is a chemoattractant for neutrophils and monocytes, thereby influencing the immune composition of tissues[50]. Thus, the environment of the Redox tumor may reduce the levels of inflammation in this subtype and prevent infiltration of immune cells.

Furthermore, we found the Redox subtype harbored significant increases in the 3q2 locus that contains *TP63*, *SOX2*, and *PIK3CA*. These increases can further amplify changes induced by *NFE2L2/KEAP1* mutations or *NFE2L2* copy number gains. SOX2

can enhance translation of oncogenic proteins driven by key cancer transcription factors, including NFE2L2, while signaling from PIK3CA can maintain NFE2L2 protein levels[51,52]. Integration of our proteogenomics results with public RNA interference and CRISPR data revealed vulnerabilities in SCC that included PSAT1 and TFRC, metabolic enzymes with direct ties to NFE2L2 as well as TP63. PSAT1 is a known target in *NFE2L2/KEAP1* mutant lung cancer cell lines and was elevated in lung cancer and PDX tissues compared to adjacent lung tissues, suggesting serine biosynthesis could be an attractive target in SCC[30,43]. The TP63 isoform, ΔNp63, transcriptionally regulates amylin, which effects glycolysis and production of reactive oxygen species. This proteoform has been already identified as a vulnerability in lung adenocarcinoma[53]. Along with a more recent report finding *GLUT8* as a vulnerability in *KRAS/KEAP1* mutant lung cancer cell lines, these results suggest metabolic vulnerabilities exist within Redox[54]. The Redox proteomic subtype showed a clear correspondence to the Wilkerson et al. mRNA groups with 84% classified as classical, and other previously published classifiers

also align with Redox[4,5]. Given the prominent proteome biology associated with detoxification and cytoprotective proteins, this SCC subtype could have intrinsic mechanisms of resistance to cytotoxic agents and radiation, which needs further examination in patient cohorts. Since *KEAP1* mutations have been suggested to play a role in therapy resistance, the Redox subtype may also be expected to be resistant to targeted agents (i.e., EGFR, FGFR, MET, and PI3K inhibitors) previously pursued in SCC.

The Inflamed subtype consisted of tumors with higher abundance of proteins reflecting neutrophil/myeloid biology and MHC proteins (Supplementary Fig. 11). Our findings suggest that this class of tumors may have strong underlying biological mechanisms that recruit various immune cell types, which is relevant to identifying combination immune therapies that eliminate the tumor's ability to evade the immune system. While the Inflamed subtype was particularly obvious from the proteomic data, similar suggestions have been made through more complex analyses of multiple data types. Integrative genomic studies by TCGA and Chen et al. suggested a combined basal and secretory subtype consistent with our Inflamed subtype[4,5].

Inflamed A had higher expression levels of proteins associated with neutrophil/myeloid biology, adhesion, and migration. Nearly half of Inflamed A samples (43%) corresponded to the basal mRNA group, which was portrayed as a cell adhesion phenotype in part due to elevated S100 family expression, but S100 protein family members are also important in neutrophil adhesion and chemotaxis[19,55]. Inflamed B had a significantly higher MHC II protein expression and corresponded to 19% of our patient population, strikingly similar to the 20% response rate of SCC patients treated with the immune checkpoint inhibitor, Nivolumab[56]. A majority of Inflamed B samples (75%) corresponded to the secretory mRNA group. *PD-1* and other immune checkpoints, but not *PD-L1*, have been shown to have elevated gene expression in the secretory group[57]. We confirmed significantly higher *PD-1* RNA expression in the Inflamed subtype, which is potentially indicative of exhausted T-cells[58], but we did not detect PD-1 or PD-L1 in our proteomics experiments. We identified significantly higher infiltration of monocytes in the Inflamed subtype (Fig. 2f), consistent with a recent observation that the secretory and basal groups are highly enriched for and driven by monocytes[59]. Proteins associated with the Inflamed subtype, especially Inflamed B associated MHC proteins, may augment our understanding of which SCCs are likely to respond to PD-1/PD-L1 immune checkpoint therapy.

In our cohort, TLN were associated with less likelihood of tumor recurrence and improved overall survival. Immunohistochemistry confirmed strong CD20 staining in TLN indicative of organized infiltrates of B-cells within our cohort. Similar TLN structures have been identified in patients receiving therapeutic HPV vaccines for cervical cancer and also in allografted kidneys undergoing rejection[25]. Further studies are critical to understand these structures and determine if treatment strategies can be developed based on their function in immune surveillance.

The third subtype, Mixed, was the least prevalent subtype and had reduced expression of both immunological and redox proteins. A smaller third group consisting of all four Wilkerson et al. classifications was also identified in previous publications[4,5]. Our findings show that proteomics can capture similar information to these multi-platform genomic studies, and taken together, these findings are supportive of the existence of this third obscure subtype of SCC. Our results suggest a role for Wnt signaling biology in this tumor (Supplementary Fig. 12), as we found elevated proteins associated with Wnt biology, enrichment of *APC* mutations, and the presence of an oncogenic mutation in *CTNNB1*.

We have produced a proteogenomic landscape that has integrated the known changes in the genome and transcriptome with proteome data; interpretation was enhanced by comprehensive clinical and outcome data. Integration of different data types and additional publicly available resources has enabled the identification of molecular subtypes, which can be further explored to implicate vulnerabilities that can be exploited using therapeutic strategies. This resource provides important insights into the biology of lung SCC and suggests that therapeutic efforts be refocused on immune combinations, including those that address tumor neutrophils, for Inflamed tumors or exploitation of metabolic vulnerabilities within *NRF2/KEAP1* mutant SCC in the Redox subtype. Because the omics and clinical data are publicly available, members of the lung cancer research community can examine additional hypotheses to further elucidate the biology of these tumors and derive additional benefits from this dataset.

## Methods

**Clinical samples**. We identified 108 patients consented to the Total Cancer Care™ protocol that had donated snap frozen lung squamous cell carcinoma tumor tissues with linked molecular data as well as clinical history, tumor pathology, and patient outcomes (Table 1, Supplementary Data 1, Supplementary Fig. 1). Samples were collected under the Moffitt's Total Cancer Care protocol (Liberty IRB # 12.11.0023) and Moffitt's General Banking protocol (IRB #: USF 101642). Described experiments were considered non-human subjects research and performed under protocol MCC #50083. De-identified clinical attributes including tumor cellularity, tumor necrosis, smoking status, and TNM staging can be found in Supplementary Data 1. Cohort demographics not included in supplementary data including age, race, and gender are available through dbGaP or by request to authors. Frozen tissue samples were randomized with respect to stage, recurrence, gender, vital status and age, were homogenized with a BioPulverizer (Biospec) in liquid nitrogen, and split into two approximately equal aliquots for genomic and proteomic analyses.

**Expression proteomics**. Homogenized tissue samples were resuspended in lysis buffer containing 20 mM HEPES, pH 8.0, 9 M urea, 1 mM sodium orthovanadate, 2.5 mM sodium pyrophosphate, and 1 mM β-glycerophosphate (Supplementary Fig. 13). After brief sonication, the lysate was cleared by centrifugation at $10,000 \times g$ at 15 °C for 30 min. Protein concentration was determined by Bradford Assay (Coomassie Plus, Pierce), and 1 mg of total protein was digested for each sample (Supplementary Fig. 14). The proteins were reduced with 4.5 mM DTT at 60 °C for 20 min followed by alkylation with 11 mM iodoacetamide at room temperature for 15 min in the dark. The sample was then diluted 4-fold to a final concentration of 2 M urea, 20 mM HEPES, pH 8.0, and trypsin digestion was carried out overnight at 37 °C with an enzyme/substrate w/w ratio of 1/25.

The digested peptide solution was acidified with 20% TFA to a final TFA concentration of 1%. After incubation at room temperature for 10 min, the solution was cleared by centrifugation at $10,000 \times g$ at 15 °C for 15 min. Sep-Pak cartridges were washed with 5 ml acetonitrile followed by 3 ml and 4 ml washes with Sep-Pak solvent A (aqueous 0.1% TFA). After acidified peptides were loaded, the cartridge was washed with 1, 5, and 6 ml of Sep-Pak solvent A. Elution was carried out three times using 2 ml of Sep-Pak solvent B (aqueous 40% acetonitrile with 0.1% TFA). After freezing, the peptides were lyophilized to dryness over 2 days. For DIA, aliquots (500 ng) of total protein digest were injected for each sample. Two aliquots (100 μg each) of total protein digest were retained for TMT labeling. The remaining material was saved for future experiments (e.g., phosphoproteomics).

The same UPLC conditions were used as described above for the TMT experiments. The mass spectrometry method utilized MS[1] scans alternated with looped 18 narrow window data independent acquisition (DIA) tandem mass spectrometry scans covering the m/z range from 450 to 1080. The MS/MS isolation windows from 450 to 900 were set at 5 Th and the isolation windows from 900 to 1080 were set at 8 Th. Resolution was set at 70,000 for MS[1] and 17,500 for MS/MS. Twenty-five femtomoles of standard peptides (PRTC) were spiked in each LC-MS/MS analysis to monitor instrument performance. The acquisition sequence can be found in Supplementary Data 3.

The tandem mass tag (TMT) experimental design, specifically how pools and tumors were assigned in 29 batches, can be found in Supplementary Data 3. After Sep-Pak, 100 μg aliquots of tryptic peptides were lyophilized overnight and dissolved in 100 μg of aqueous 100 mM Triethylammonium bicarbonate (TEAB) buffer. TMT reagents (0.8 mg) were equilibrated to room temperature and 41 μl of anhydrous acetonitrile was added to each tube and vortexed for 5 min. The TMT labeling was carried out by mixing the TMT reagent with peptide mixture and incubating at room temperature for 1 h. The labeling was quenched by adding 8 μl of 5% hydroxylamine and incubating for 15 min. For quality control, an aliquot (1 μl) of the labeled digest was analyzed by LC-MS/MS to determine the effectiveness of the labeling.

After chemical labeling, LC-MS/MS analysis was performed on each TMT channel followed by Mascot/Sequest searches, and the results were summarized in Scaffold. Spectral counting was used to calculate percentage of peptides labeled

with TMT reagent (Supplementary Data 3). One tumor sample (batch 05, channel 126) required relabeling. For two experiments, the labels for channel 128 and channel 129 were switched (batch 24 and batch 29), and the tumor identification numbers were corrected prior to downstream analysis. In addition, MaxQuant analysis was also performed on each TMT channel to verify correct TMT channel was labeled for each sample and to examine the crosstalk between channels[60] (Supplementary Data 3).

Dried peptides were reconstituted in 20 mM ammonium formate, pH 10. bRPLC separation was carried out on Dionex Ultimate3000 RSLC UPLC using a 4.6 mm ID × 100 mm column (XBridge, Waters) packed with C18, 3.5 μm particle size at a 0.6 ml/min flow rate. The gradient setting was: 100% bRPLC A (aqueous 2% acetonitrile with 5 mM ammonium formate, pH 10) for 9 min, then the concentration of bRPLC solvent B (aqueous 90% acetonitrile with 5 mM ammonium formate, pH 10) was increased to 6% over 4 min, to 28.5% over 50 min, to 34% over 5.5 min, and to 60% over 13 min, and then kept constant for 8.5 min prior to re-equilibration at the original conditions. Twelve concatenated fractions were analyzed with LC-MS/MS for each TMT experiment (Supplementary Fig. 15).

A nanoflow ultra-high performance liquid chromatograph (RSLC, Dionex, Sunnyvale, CA) interfaced with an electrospray quadrupole-orbital ion trap mass spectrometer (Q Exactive Plus, Thermo, San Jose, CA) was used for tandem mass spectrometry peptide sequencing experiments. The sample was first loaded onto a pre-column (2 cm × 100 μm ID packed with C18 reversed-phase resin, 5 μm particle size, 100 Å pore size) and washed for 8 min with aqueous 2% acetonitrile and 0.04% TFA. The trapped peptides were eluted onto the analytical column, (C18, 75 μm ID × 50 cm, 2 μm, 100 Å, Dionex, Sunnyvale, CA). The 90 min gradient was programmed as: 95% solvent A (aqueous 2% acetonitrile + 0.1% formic acid) for 8 min, solvent B (aqueous 90% acetonitrile + 0.1% formic acid) from 5% to 38.5% in 60 min, then from 38.5 to 90% in 7 min, and held at 90% for 5 min, followed by solvent B from 90 to 5% in 1 min and re-equilibration for 10 min. The flow rate on analytical column was 300 nl/min. Sixteen tandem mass spectra were collected in a data-dependent manner following each survey scan using 60 sec exclusion for previously sampled peptide peaks using normalized collision energy 30. Twenty-five femtomoles of PRTC was spiked in each LC-MS/MS analysis to monitor instrument performance. Total LC-MS/MS time was 87 days. The acquisition sequence can be found in Supplementary Data 3 (Supplementary Fig. 16).

**Molecular genomics.** Qiagen's AllPrep DNA/RNA Mini kit was used for the simultaneous purification of genomic DNA and total RNA from the homogenized tumor tissue aliquot.

RNAseq was performed using the NuGen Ovation Encore Complete RNAseq kit, which generates strand-specific total RNAseq libraries (Nugen, Inc., San Carlos, CA). Following quality control screening on the NanoDrop to assess 260 nm/230 nm and 260 nm/280 nm light absorbance ratios as a metric for sample purity, the samples were screened on the Agilent BioAnalyzer RNA Nano chip to generate an RNA Integrity Number (RIN) (Agilent Technologies, Santa Clara, CA). An aliquot of DNase-treated total RNA (100 ng) was then used to generate double-stranded cDNA, which was initiated with selective random priming allowing for the sequencing of total RNA, while avoiding rRNA and mtRNA transcripts. After primer annealing at 65 °C for 5 min, a first strand cDNA synthesis reaction was performed at 40 °C for 30 min using kit-supplied reverse transcription reagents (Nugen). Second-strand cDNA synthesis was performed in a 70 μl reaction volume at 16 °C for 1 h and the reaction was stopped by adding 45 μl of stop solution. The double-stranded cDNA was then fragmented to ~200 bp with the Covaris M220 sonicator (Covaris, Inc., Woburn, MA), followed by purification with Agencourt RNAClean XP (Beckman Coulter Life Sciences, Indianapolis, IN). The fragmented DNA was suspended in 10 μl of nuclease-free water, and end-repair was performed in a 13 μl volume for 30 min at 25 °C, followed by a heat inactivation at 70 °C for 10 min. A sample-specific indexed adapter was ligated to the end-repaired DNA for 30 min at 25 °C, followed by strand selection, a 1.8X volume RNAClean XP bead purification, and a second round of strand selection. Thirteen cycles of library amplification followed by a 1.2X volume RNAClean XP purification of the strand-selected library was performed, and finally, the library DNA was resuspended in 30 μl of nuclease-free water.

Final libraries were screened for library fragment size distribution using an Agilent BioAnalyzer High-Sensitivity DNA Chip. Libraries were then quantitated using the Kapa Library Quantification Kit (Roche Sequencing, Pleasanton, CA), normalized to 4 nM, and then sequenced on an Illumina NextSeq 500 150-cycle high-output flow cell in order to generate ~100 million paired-end reads of 75-bases in length for each sample (Illumina, Inc., San Diego, CA).

In order to assess the somatic mutation status in the squamous cell carcinoma samples, a custom Agilent SureSelect panel covering 154 total genes (151 genes from the ClearSeq Comprehensive Cancer Panel plus KMT2D, KEAP1, and NFE2L2) was designed and used for the enrichment of DNA libraries generated using the 200 ng input protocol of the Agilent SureSelect XT Library Kit (Agilent Technologies, Santa Clara, CA).

Genomic DNA samples were qualitatively assessed using the Agilent TapeStation and Genomic DNA screentape and the Qubit dsDNA high-sensitivity assay was used to quantify the samples (ThermoFisher, Waltham, MA). The samples were diluted in nuclease-free water to a concentration of 4 ng/μl at a

volume of 50 μl and fragmented in 50 μl AFA Fiber screw-cap microtubes using a Covaris M220 sonicator (Covaris, Woburn, MA). The M220 Covaris instrument settings to fragment DNA to a target size of 150 to 200 bp were: Duty Factor (10%), Peak Incident Power (175), Cycles per Burst (200), Treatment Time (360 s), and water bath temperature (4 °C).

The fragmented DNA samples were then qualitatively assessed using a high-sensitivity DNA chip on the Agilent BioAnalyzer, and the fragmented DNA samples with an average size of 150–200 bp were then processed using the Agilent SureSelect XT library kit. The fragmented DNA samples were end-repaired using Klenow and T4 DNA polymerases along with a T4 polynucleotide kinase at 20 °C for 30 min. The blunt-end product was purified using Ampure XP beads with a 1.8X bead slurry to sample ratio (Beckman Coulter Life Sciences, Indianapolis, IN). Next, the 3' ends of the blunt-end products were adenylated using dATP and Klenow fragment without exonuclease activity at 37 °C for 30 min. Following the adenylation reaction, another 1.8X ratio bead purification was performed. The adenylated product was then ligated to an Illumina sequencing compatible paired-end adapter using T4 DNA ligase at 20 °C for 15 min. The ligated product was purified using Ampure XP beads with a 1.8X ratio and the adapter-tagged purified product was then amplified using the Agilent Fusion Pfu-based DNA polymerase. An adapter-specific forward primer and an Illumina indexing reverse primer provided by Agilent were used for amplification. The cycling parameters were as follows: 98 °C for 2 min, 10 cycles of the following temperature program (98 °C for 30 s, 65 °C for 30 s, and 72 °C for 1 min), followed by a final elongation step for 10 min at 72 °C. The amplified product was purified using Ampure XP beads with a 1.8X ratio.

The DNA libraries were assessed for size and quantity using the DNA 1000 on the BioAnalyzer, and each DNA library was verified to have an average peak between 225 and 275 bp. The concentration generated from the QC report was used to calculate the input into the hybridization and capture reactions. In preparation for the hybridization and capture reactions, 750 ng of each DNA library was concentrated to a volume of 3.4 μl, and hybridization buffer, a SureSelect blocking mix, an RNase blocking mix, and a capture library hybridization mix were prepared. The blocking mix (5.6 μl) and DNA library (3.4 μl) were mixed by pipetting, carefully sealed, and transferred to a thermal cycler running the following program: 95 °C for 5 min, 65 °C for 16 h. After 5 min at 65 °C, the thermal cycler was paused and 20 μl of the capture library hybridization mix was added to the denatured DNA library while on the thermal cycler. The combined mixtures were mixed quickly by pipetting and carefully re-sealed before resuming the incubation at 65 °C for 16 h.

In preparation of the capture purification, streptavidin-coated beads were washed three times using a binding buffer and a magnetic separation device and a water bath was set to 65 °C. The enriched target DNA was captured using 200 μl of washed streptavidin-coated magnetic beads and mixed on a Jitterbug at 1600 rpm for 30 min at room temperature (Boekel Scientific, Trevose, PA). The beads in each sample were then pulled down by the magnetic separator and washed with 200 μl of SureSelect Wash Buffer 1 and re-captured for 15 min. The target-bound beads were washed three times at 65 °C with SureSelect Wash Buffer 2 and resuspended in 30 μl of nuclease-free water.

The enriched DNA libraries were PCR amplified and indexed using sample-specific and universal primers. Following a 2 min 98 °C denaturation, 16 cycles of PCR were performed as described above except that 57 °C was used as the annealing temperature. The amplified enriched libraries were Ampure XP purified with a 1.8X ratio and suspended in 30 μl of nuclease-free water. After size assessment on the high-sensitivity D1000 TapeStation DNA assay, the samples were quantified by Qubit and with the quantitative PCR-based Kapa Library Quantification kit (Roche Sequencing, Pleasanton, CA). Samples were diluted to 4 nM, pooled and prepared for sequencing on the NextSeq 500 sequencer. Two paired-end 2 × 75 v2 mid-output sequencing runs were performed in order to generate a target coverage of >150X of the genes in each sample.

In order to assess CNV and loss-of-heterozygosity (LOH) status, the Affymetrix CytoScan HD Assay was performed. The CytoScan HD assay uses approximately 750,000 SNP probes and 1.9 million non-polymorphic probes to report genome-wide copy number aberrations at 25–50 kilobases in size.

Starting with 250 ng of tumor-derived DNA diluted at 50 ng/μl, an Nsp I digestion was performed at 37 °C for two hours in the presence of 1X bovine serum albumin (BSA). Following heat inactivation at for 20 min at 65 °C, vendor-supplied Nsp I adaptors were ligated to the digested sample DNA for three hours at 16 °C, followed by heat inactivation at 70 °C for 20 min. The ligated DNA samples were diluted four-fold and ligation-mediated PCR was performed in quadruplicate using an Affymetrix-specific TITANIUM Taq PCR kit (Clontech Laboratories, Inc., Mountain View, CA) following the CytoScan Assay User Manual.

After 30 cycles of PCR amplification, 3 μl of the amplified product was screened on a 2% agarose gel, and the PCR replicates were pooled, bead-purified, and quantitated according to the protocol. Amplified DNA samples were DNase I fragmented at 37 °C for 35 min, and the DNase was heat inactivated at 95 °C for 15 min. Two μl of fragmented PCR product was screened on a 4% agarose gel, and the fragmented DNA was end-labeled by terminal deoxynucleotidyltransferase in the presence of biotin for four hours at 37 °C followed by heat inactivation at 95 °C for 15 min.

A hybridization buffer was prepared and added to the fragmented and labeled DNA; after a 10 minute 95 °C denaturation, CytoScan HD arrays were hybridized

at 50 °C for 16 h in an Affymetrix GeneChip Hybridization Oven 640. The following day, the hybridized arrays were washed and stained with biotinylated anti-streptavidin and a streptavidin-R phycoerythrin (SAPE) conjugate on the Affymetrix GeneChip Fluidics Station 450 according to the protocol. Finally, the arrays were scanned on an Affymetrix GeneChip Scanner 3000 7 G with autoloader and inspected for any chip defects or artifacts. The raw data generated from the assay were normalized, copy number status was calculated, and the data were reviewed for quality using the Affymetrix Chromosome Analysis Suite (ChAS) v. 3.1.

**RNAseq**. Sequence reads were aligned to the human reference genome (hs37d5) in a splice-aware fashion using Tophat2[61], allowing for accurate alignments of sequences across introns. Aligned sequences were assigned to exons using the HTseq package against RefSeq gene models to generate initial counts by region[62]. Normalization, expression modeling, and difference testing were performed using DESeq2[63]. RNAseq quality control includes RSeQC to examine read count metrics, alignment fraction, chromosomal alignment counts, expression distribution measures, and principal component analysis (PCA). PCA and hierarchical clustering were used to examine the data for outliers and identify any concerns about instrument performance or experimental design prior to further analysis[64]. RNASeq abundances were $\log_2$ transformed, then de-batched with COMBAT to correct for kit-related batch effect (kit lot #: 1511366-B, 1412465-C, 1604466)[65]. We were able to quantify expression for 19,559 genes across the entire cohort.

**Targeted exome sequencing**. Sequence reads were aligned to the reference human genome (hs37d5) with the Burrows-Wheeler Aligner (BWA)[66], and duplicate identification, insertion/deletion realignment, quality score recalibration, and variant identification were performed with PICARD (http://broadinstitute.github.io/picard/) and the Genome Analysis ToolKit (GATK)[67]. All genotypes (even reference) were determined across all samples at variant positions using GATK. We also sequenced 12 normal blood samples using the same procedures in order to remove artifacts and other false positives common to both tumor and normal samples[68]. Various quality control measures were applied to determine depth of coverage in each sample across the targeted genes. Sequence variants were annotated using ANNOVAR[69], and summarized using spreadsheets and a genomic data visualization tool, VarSifter[70]. Additional contextual information was incorporated, including allele frequency from other studies such as 1000 Genomes Project and the NHLBI Exome Sequence Project, in silico functional impact predictions, and observed impacts on function from databases such as ClinVar[71].

**Copy number analysis**. The Affymetrix CytoScan HD microarray was used for the identification of CNVs and Chromosome Analysis Suite (ChAS) software was utilized for analysis. The array consists of 2,696,550 probes that include 743,304 SNPs and 1,953,246 non-polymorphic probes at an average spacing of 1 probe per every 800 bp throughout the entire human genome. The average spacing of sampling for RefSeq genes is 1 probe per 880 bp and 96% of genes are represented. To infer ancestry of the samples, we used 395 HapMap samples (350,000 + common SNPs with >95% call rates) as reference for four ethnically diverse populations. SNP array data quality was assessed with Affymetrix "Median of the Absolute Values of all Pairwise Differences" (MAPD), which estimates the variability of $\log_2$ ratios over the complete array with robustness to overcome high biological variability as frequently found in tumor DNA samples. MAPD values below 0.25 were considered to be indicative for good quality. To calculate the CNV, the data were normalized to baseline reference intensities using 270 HapMap samples and 90 healthy normal individuals included in ChAS. The copy number states were determined by the Hidden Markov Model (HMM) algorithm using the human genome (hg19/NCBI build 37). To prevent the detection of false-positive CNVs arising due to array noise, only alterations that involved at least 50 consecutive probes and alterations >400 kb in length were considered in the analysis of gains or losses. Amplifications and deletions were analyzed separately. The detected CNVs were evaluated separately in terms of frequency and length.

**LC-MS/MS processing**. Protein identification was performed using the RefSeq human protein sequence database (version 78) supplemented with bovine and porcine trypsin sequences. Thermo.RAW files were converted to mzML peaklists using ProteoWizard msConvert[72]. We used three separate search engines, MyriMatch (version 2.2.10165)[73], MS-GF + (version 20160629)[74], and Comet (version 2016.01rev1)[75] to assign peptide sequences to tandem mass spectra. Precursor tolerance was set at 10 p.p.m. and fragment tolerance at 0.5 $m/z$, allowing semi-tryptic peptides using forward and reverse peptide sequences. Static modifications of 229 Da on lysine and N-terminal (TMT label) and 57 Da on cysteines (carboamidomethylation) were specified; while dynamic modifications of 16 Da on methionine (oxidation) and −17 Da on N-terminal Glutamine residues (pyroglutamine) were included.

**Protein assembly**. Spectral identification files from Myrimatch, MS-GF + and Comet (2088 files total) were converted to IDPicker 3 index files (idpXML) using IDPicker 3 (www.idpicker.org) and summarized in a single IDPicker database file[76]. This assembly was filtered at a stringent 0.1% peptide-to-spectrum (PSM)

FDR while requiring a minimum of 2 distinct peptides per protein, as described previously[10]. To increase the number of high-quality PSMs, we limited the assembly to this set of proteins while relaxing the PSM to 1%. This process resulted in the identification of 2,954,487 spectra representing 158,160 distinct peptides corresponding to 8300 protein groups with a protein FDR of 4.75%. We further limited the number of proteins to those observed in at least 10% of the tumors, which resulted in a total of 4880 protein groups with a protein FDR of 1.3%. TMT reporter ion intensities were normalized across each peptide to correct for mixing variability and exported from IDP3 for further processing.

We used a method similar to internal reference scaling to derive protein group expression values that were comparable across TMT 6-plex experiments[77]. Each of the 29 TMT 6-plexes contained four tumors and two tumor replicates, totaling 174 samples (116 tumors, 58 pool replicates). The pools were assayed on every TMT 6-plex to allow for controlling for variability between TMT experiments, with one pool fixed in channel 126 and the other varying channel between TMT experiments. Within each 6-plex injection replicate, spectral abundances for each channel were normalized with IRON (using the console command iron_generic --proteomics) against the m/z 126 reporter ion (ch-126 pool)[78]. For each spectrum, log ratios were calculated vs. the ch-126 pool. Protein-level rollup of log ratios ($log\ ratio_{protein}$) were calculated by averaging the individual spectral log ratios within each protein group. Initial individual ch-126 protein-level abundance estimates were calculated as the geometric mean of the unlogged spectral abundances within each protein group.

Protein-level abundance estimates for all individual ch-126 pools were then IRON normalized against the median ch-126 pool ("TMT-126 TMT13_01"; Supplementary Data 3) across all injection replicates. Abundance scales ($Scale_{protein}$) for each protein group were then calculated as the geometric mean of the normalized unlogged ch-126 protein-level abundances. Normalized protein-level abundances for each injection replicate were then calculated by multiplying the protein-level abundance scales times the protein-level ratio [$Scale_{protein} \times \exp$ ($log\ ratio_{protein}$)]. Abundances were then $\log_2$ transformed and injection replicates averaged to yield the dataset used for all further downstream proteomics analyses.

**Consensus clustering**. The top 1000 most variable proteins by median absolute deviation were clustered using Consensus Cluster Plus with missing values ignored (treated as "NA")[14]. Consensus Cluster Plus parameters were pItem = 0.8, pFeature = 1, clusterAlg = "hc", distance = "pearson", seed = 1234, innerLinkage = "complete", finalLinkage = "complete", and corUse = "pairwise.complete.obs". Consensus clustering output provides cluster assignments and stability evidence for the identified k clusters. The process is repeated for multiple values of k. The purpose of the CDF plot (Supplementary Fig. 4A) is to find the smallest k at which the change of the area under the CDF curve (Supplementary Fig 4B) is small, suggesting that there is not much benefit to further divisions. Here, we chose k = 5 since it was the first point which corresponded to less than 10% change in area, and larger k would have only given small additional increases in area. For our RNAseq-based assignments, we followed a similar strategy and took the top 1000 most variable genes by median absolute deviation (also observed in >90% of the samples) and clustered with identical parameters.

**Gene-based sample classification**. Wilkerson et al. predictor centroids were used to classify our RNAseq expression data according to the Wilkerson classification scheme (classical, basal, secretory, and primitive)[6].

Unless otherwise noted, missing values were ignored (treated as "NA"). We tested for differential expression of genes/proteins using a nonparametric Wilcoxon rank-sum tests, and we defined significantly differentially expressed as having at least ±1.5 fold-change and a Benjamini-Hochberg adjusted P-value ($P_{adj}$) less than or equal to 0.05. Unless otherwise noted, these comparisons and others were made for a given subtype or group vs. the rest of the cohort. For mutation and copy number comparisons, we report the results of two-sided Fisher exact test P-values for a given group compared to the rest of the population unless otherwise noted. We used Spearman's correlations for correlating copy number, RNA, and protein expression with the "pairwise.complete.obs" argument in R, and we required at least 10% non-missing values in both sets of values being correlated. Pathway enrichment was assessed using Enrichr and MSigDB[16,17]. Principal component analysis utilized the 4880 protein groups that were observed in >90% of the samples with missing values imputed to 0.

**Deconvolution approaches**. The ESTIMATE R package was used to calculate immune infiltration on RNAseq expression data[18]. RNAseq data were unlogged and missing values imputed with 0 for the purpose of running CIBERSORT to assess relative abundance of immune cell types[7]. CIBERSORT was run in relative mode with the LM22 signature gene file, 100 permutations, and quantile normalization disabled. Seventy five tumors passed the CIBERSORT goodness of fit threshold P-value < 0.05 for subsequent analysis. xCell was run using the default "N = 64" gene signature with the "RNA-seq?" option selected[20].

**Gene and protein nomenclature**. We refer to genes and their protein products by the official symbol provided by the Human Genome Organization's Gene Nomenclature Committee (HGNC). If a gene's synonym is more commonly used,

for example "NRF2" instead of the official "NFE2L2", then we provide both. We have referred to protein groups consisting of two or more similar gene symbols (e.g., "DEFA1 DEFA1B") as the first entry (e.g., "DEFA1") to improve readability and avoid confusion. Complete protein group information including naming is provided in proteomics Supplementary Data 3.

**Survival analysis**. Cox regression for overall survival (OS) and recurrence-free survival (RFS) was fitted to gene-level CNV, RNAseq expression, and protein expression data. Time-to-event for OS was time in years from date of sample collection to date of death and censored at date of last contact, and time-to-event for RFS was time in years from date of sample collection to date of first clinically confirmed recurrence if a patient recurred or died and censored at the date of last contact. The event for OS was death and for RFS was recurrence or death.

**Meta-analysis**. Meta-analysis was performed using the metaphor R package[79] on gene-level CNV, RNAseq expression, and protein expression data. Three thousand four hundred eighty-four genes and their corresponding proteins present in all three datasets were used. When mapping gene symbols across datasets, in the few cases of duplicated gene symbols in protein expression data (due to protein grouping), one gene (protein group) was arbitrarily chosen to be included. A meta-analytic random-effects model was fitted using empirical Bayes as the heterogeneity estimator. The Storey q-value method was used to adjust p-values, and 15 genes were found with a q-value less than or equal to 0.3. Hazard ratios with 95% confidence intervals were plotted, and the upper limit of the confidence intervals were truncated at 3.5 for better visualization.

**Pathology**. Tertiary lymph nodes (TLN) were scored using H&E slides from our patient tumor samples (Supplementary Data 1). TLN scoring was 0—no TLN identified, 1—few TLN identified, 2—many TLN identified, 3—a single outlier sample with so many TLN that it was given its own score.

CD20 Immunohistochemistry assays (IHC) were run on the automated Ventana Discovery XT platform (Supplementary Data 1). Anti-CD20 (Ventana 760–2531), a mouse monoclonal (L26) was used at predilute concentration (0.3 µg/ml) with Cell Conditioning 1 solution for antigen retrieval along with a heated 16 min primary incubation. OmniMap anti-Ms-HRP for 16 min was used for secondary incubation. Slides were counterstained with Ventana hematoxylin (760–2021) and Bluing Reagent (760–2037) for 4 min each. Ventana Ms IgG (760–2014) was used as the mouse IgG control and run under same conditions. Human tonsil tissue sections were used as both a positive and Ms IgG control.

CD8/CD33 was run as dual IHC stain on the automated Ventana Discovery XT platform (Supplementary Data 1). Anti-CD8 (Ventana 790–4460, predilute, 0.3 ug/ml) a rabbit monoclonal (SP57) was diluted 1:15 with Dako antibody diluent (S0809) to a final concentration of 0.02 ug/ml and anti-CD33 (Leica PA0555, predilute, 10 mg/ml), a mouse monoclonal (PWS44). The combined antibodies used Cell Conditioning 1 for antigen retrieval and a primary antibody heated incubation time of 40 min. Anti-CD8 was visualized using OmniMap DAB and anti-CD33 was stained with Ultra Map-Red. OmniMap DAB secondary incubations were reacted for 4 min while UltraMap Red for 16 min. Ventana (Rb IgG) (760–1029) was used as the rabbit IgG control and Ventana (Ms IgG) (760–2014) was used as the mouse IgG control and run under same conditions. Human tonsil tissue sections were used as the positive control for CD8/CD33 staining.

The scoring of CD33+ intratumoral neutrophils (T-CD-33); and CD33+ stromal neutrophils (S-CD-33) were based on criteria for semiquantitative assessment of TILs, described by Schalper et al. (Supplementary Data 1)[80]. A score of 0 indicated the virtual absence of positive neutrophils in evaluation of either T-CD-33 or S-CD-33; a score of 1+ indicated a relatively low amount of T-CD-33 compared against all nucleated cells in the tumor area or S-CD-33 compared against all nucleated cells in the stromal area (<30%); a score of 2+ corresponded with a relatively moderate amount of positive neutrophils (30–60%); and a score of 3+ corresponded with a high amount of CD33+ neutrophils (>60%).

**Reporting summary**. Further information on research design is available in the Nature Research Reporting Summary linked to this article.

## Data availability

Proteomic data have been uploaded to PRIDE (https://www.ebi.ac.uk/pride/archive/). The DIA QC data have been deposited under the accession code PXD010357 (https://doi.org/10.6019/PXD010357). The TMT data have been deposited under the accession code PXD010429 (https://doi.org/10.6019/PXD010429). Genomic and transcriptomic data were uploaded to dbGaP with the accession code phs001781.v1.p1. Cohort demographics not included in supplementary data including age, race, and gender are available through dbGaP or by request to authors.

## Code availability

The code used for the primary analyses can be found here: https://github.com/pstew/proteogenomics_scc.

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

## Acknowledgements

This work was supported by the Moffitt Lung Cancer Center of Excellence as well as the Biostatistics and Bioinformatics, Proteomics, Molecular Genomics, Analytic Microscopy, Collaborative Data Services, and Tissue Shared Resources at the H. Lee Moffitt Cancer Center & Research Institute, an NCI-designated Comprehensive Cancer Center (P30-CA076292). P.A.S. thanks the National Institutes of Health Loan Repayment Programs for an award that supported him during this work. P.A.S. warmly thanks Brooke Fridley, Richard Reich, and John Schatzle for their continued support. We acknowledge the help of Marek Wloch with nucleic acid extraction, Noel Clark with immunohistochemistry, Jonathan Nguyen and Joseph Johnson with image capture and analysis, Erin Siegel with clinical outcome data collection, Myra George with Total Cancer Care™ data management, Susan Sharpe with public data submission, and Melissa Avedon with project management. We would like to thank Elsa Flores for answering many questions about TP53 family members.

## Author contributions

Conception and design: P.S., J.K., S.E. and E.B.H. Development of methodology: P.S., E.W., G.Z., S.Y., B.F., R.S., T.A.B., J.K., J.T., A.C., S.E. and E.B.H. Acquisition of data (provided animals, acquired and managed patients, provided facilities, etc.): P.S., S.Y., B.F., V.I., T.M., C.Z., J.K., E.B.H. Analysis and interpretation of data (e.g., statistical analysis, biostatistics, computational analysis): P.S., E.W., R.S., M.C., L.C., F.P., Y.Z., Z.C., C.C., K.M.F., J.M.F., J.J.S., G.M.D., A.A.B., T.A.B., J.K., J.T., A.C., S.E., E.B.H. Writing, review, and/or revision of the manuscript: P.S., E.W., G.Z., S.Y., B.F., R.S., M.C., V.I., T.M., C.Z., L.C., F.P., Y.Z., Z.C., C.C., R.T., Z.T., K.M.F., J.M.F., J.J.S., A.A.B., T.A.B., J.K.,

J.T., A.C., S.E., E.B.H. Administrative, technical, or material support (i.e., reporting or organizing data, constructing databases): P.S., E.W., S.Y., B.F., R.S., M.C., V.I., T.M., C.Z., L.C., F.P., Y.Z., Z.C., C.C., R.T., Z.T., K.M.F., J.M.F., J.K., J.T., S.E., E.B.H. Study supervision: E.B.H. Other (pathology support): T.A.B., J.J.S.

## Additional information

**Competing interests:** J.K.'s laboratory was funded by a sponsored research agreement from Proteome Sciences, who manufactures TMT reagents for Thermo distribution. The remaining authors declare no competing interests.

