## [Peer Review File · Nature Communications]

Reviewers' comments:

Reviewer #1 (Remarks to the Author):

Stewart et al present a comprehensive proteogenomic study of squamous cell lung cancer (SCC) including DNA copy number, somatic mutations, RNAseq and proteomics data of 108 cases. They describe three proteomic subtypes which they call "inflamed", "redox" and "wnt/stromal" that partially overlap with transcript-based subtypes. While signatures of inflammatory cells were observed in the inflamed subtype the redox subtype harbored mutations in the genes NFE2L2/KEAP1 and copy number increases in TP63, SOX2 and PIK3CA. No correlations of proteomic subtypes with survival were found, yet by histological examination tertiary lymph node features were found that correlate with good prognosis. However, no signals of these tertiary lymph nodes were explored/found in the proteomics or RNAseq data of matching tumor specimens. This aspect should be either further analyzed or if no correlation can be found further discussed.

In general, proteomics data can complement genomics data by increasing the specificity for calling tumor associated genes/pathways that overlap between both data types or by highlighting additional features that are not visible in the genomics data. The protein to RNA comparisons could be further explored, such as by a subtype-focused analysis of how SCC-related transcription factor target genes of the transcription factors SOX2, p62 and NFE2L2 are regulated on RNA and protein level. SCC subtypes have been previously associated with target genes for SOX2 and p62 and these disease-related gene activities may be highlighted with better specificity using both RNA and protein data. Since one of the candidate disease genes, KEAP1, is part of an E3 ligase, it would be interesting to test if there are any post-translational regulatory events. This could be addressed by a protein-protein correlation analysis of KEAP1 to other proteins in KEAP1 wt tumors.

In summary, the inflamed and redox proteomic subtypes are very interesting findings and this study provides a useful resource dataset with a coverage of nearly 5,000 proteins across a sufficiently large cohort of >100 SCC samples. While the introduction and the discussion are very well written, the results section and main figures need to be shortened and focused on the key observations. Figure 2 can be moved to supplements and the concept pathway figures be reduced in size.

The row numbers were cut for each pdf page, therefore no row numbers are addressed in this report.

Major points of concern:

Figure 1:

- Put NFE2L2 and KEAP1 next to each other to make it easier to see mutually exclusive mutation events
- How were proteins and pathways selected? Did the authors use any FDR control for calling enriched pathways? Table S14 shows for these hits FDR controlled pvals, but no threshold are indicated here
- How do RNAs and proteins correlate for these pathways?

Page 6: The mass spectrometry MS/MS-based protein FDR at 4.75% is rather high. Is there a reason why no conventional FDR<1% was used? The subset of 4,880 proteins used for proteogenomic analysis has the same FDR?

Page 8: "Finally, because of the relevance to immune checkpoint therapy, we also examined expression of PDCD1 (PD-1) and CD274 (PD-L1). This was performed using RNAseq data, as these were not reliably quantified in the proteomics experiments."

Explain why these proteins were not reliably quantified and if those could have been excluded by using a more stringent protein FDR of less than 0.01 for protein identification in first place

Figure 4: The finding that tertiary lymph nodes are more frequent in the inflammatory proteomic

subtype is very interesting and that TLN correlate with good prognosis. Do the authors know if any of their samples for proteomic analysis contained TLN with high Bcell counts? Can any of these TLN signals be identified in RNAseq or proteomics data from corresponding tumor specimens?

Page 11: "Top elevated proteins of the combined Redox subtype included 5 AKR family members (Table S28). AKRs are indicative of NFE2L2 (NRF2) activity and are overexpressed in both SCC and lung ADC"

Indicate if top elevated proteins in redox subtype are also elevated on transcript level. Since KEAP1 is part of an E3 ligase complex it is interesting to test if redox subtype-based protein features are regulated post-translationally or driven by transcription factor-based gene expression programs.

Page 12: "We started with 303 well-correlated genes across the entire cohort (correlation of CN to Protein > 0.5 and correlation of RNA to protein > 0.5; Figure 5D; Table S11) and examined these for intersections with 94 unique proteins significantly elevated in the Redox subtype. Next, we mined redundant siRNA activity data from Project DRIVE and identified 1,101 gene knockdowns that significantly impacted at least one SCC cell line (RSA score < -3; Table S21)"

Instead of using arbitrary cut offs across thousands of proteins the authors should use here a statistical approach with significance calling for the correlations. For siRNA screens cut-off values have been described frequently in the field. For the positively scoring cell lines the authors should describe if those cell lines resemble the protein or RNA expression features of redox SCC tumors.

Minor points of concern:

Table S01, sheet "Data": include RNA- and protein-based subtype information
Do proteomic subtypes correlate with gender, smoking status or drug treatment?

Page 6: Indicate if SCC tumor collection included samples that were treatment naïve or collected after drug treatment. Also indicate if SCCs studied here were primary tumors or if some of those could be metastasis of head and neck squamous cell carcinomas.

Figure S02: This CNA heatmap would be more informative if the information was not aggregated across all patients, but if all patients were shown as columns and the chromosome regions as rows. Organize the patients by proteomic subtypes.

Page 6: Description of content of supporting data tables S03-S08 and S22-24 is missing.

Page 8: "Inflamed A, Inflamed B expressed significantly higher levels of 11 MHC Class II proteins and 9 cathepsins"

Describe statistical test used and how significance was assessed.

Table S07: TMT metadata table should also contain information about the RNA and proteomic subtypes to help understand if there any subtypes overlapping with TMT6-plexes, which can be a source of batch effects

Table S11: Add p-values and FDR corrected values for correlations, that quality of correlation can be better assessed

Table S13: How were those top proteins per subtype selected? Was any validation attempted by immunohistochemistry?

Table S26, column H: the trypsin amount should read micro-gram instead of milligram

Figure 2: RNA-protein correlation similar to previous proteogenomic studies. Move to supplements.

Page 7: Does the mixed/stromal subtype show less tumor cellularity?

Page 7: For RNAseq to protein comparison use the term transcript-protein pairs instead of gene-protein pairs

Page 8: Describe in text what supporting data tables S09-S11 and S12-S15 contain

Page 10: "We identified 15 such proteins ($q \leq 0.3$; Figure S07), suggesting new prognostic genes and proteins associated with disease recurrence. We identified high expression of the tyrosine phosphatase PTPN12 associated with poor survival which functions by counteracting oxidative stress and promoting cell survival and metastasis in breast cancer 33."

Please explain if q-val or FDR was here selected to be $<30\%$? If that is the case these findings should not be highlighted in main text. PTPN12 has been also described as a tumor suppressor counteracting receptor tyrosine kinases for example in triple negative breast cancer. Describe why PTPN12 acts in this context as an oncogene?

Are there any disease outcome related proteins that correlate with the TLN status of SCC samples?

Page 11: "We took samples with either a NFE2L2 or KEAP1 mutation (41 samples total; 32 Redox, 6 Inflamed, 3 Mixed), and when compared to wild type tumors, we found that mutated tumors were enriched for oxidative stress and metabolism pathways (Figure 5C)."

This analysis strongly overlaps with the redox subtype. Do the associated proteins have antioxidant response elements in their respective genes and constitute NFE2L2 target genes?

No attempt was described if any tumor-specific single amino acid variants were detected on protein level.

Reviewer #2 (Remarks to the Author):

In the manuscript by Stewart et al, entitled "Proteogenomic Landscape of Squamous Cell Lung Cancer", the authors detail an impressive single institution effort to characterize lung squamous tumors based on several -omic platforms. The generated data is integrated and then several approaches are taken to better understand the various biologic phenomena of different proteomic subsets. The authors are to be commended for their effort given its scope and the resource it will create for the field. Although this work will be interest to the field, some aspects are often descriptive and, in some cases, not enough has been done to validate the findings.

General Comments:

1. In the Inflamed A+B subtypes, the authors find enrichment for markers of neutrophils and their degranulation. The authors are encouraged to perform IHC and score at least a representative subset of tumors from each cluster and assess for this.
2. Several of the observations are consistent with prior work already in publication. While this is important since it comes from an independent cohort, incorporates proteomics (which none others have) and adds rigor to the field, the authors are encouraged to cite and, in some cases (per below, point #c), expound on such instances.
 - a. Higher PDCD1 (and several other immune checkpoints) but not CD274, have been shown by RNAseq to be higher in the Secretory subtype of lung squamous (Faruki et al, Journal of Thoracic Oncology, 2017)
 - b. Enrichment of the Inflamed A+B subtypes, which largely consist of Secretory and Basal subtypes, have recently been shown to be highly enriched for and driven by monocytes (Porrello et

al, Nature Communications, 2018)

c. In Korean lung squamous cancers (Seo et al, Cancer Immunology Research, 2018), they also characterize an inflammatory subtype (Subtype B in their case), that similarly enriches for Secretory and Basal subtypes. This study looked at somatic copy number variations and found these are inversely correlated with Subtype B. Along these lines, the authors are encouraged to assess whether SCNVs have similar inverse correlations with their Inflamed subtypes.

3. Please elaborate more on how the meta-analysis was performed using CNA, mRNA and protein data to relate with patient outcome. Were these the top 15 survival hits for all available loci/mRNA/proteins? Along these lines, since CNA and RNAseq data and clinical outcome are publicly available with TCGA, the authors are encouraged to assess whether any of the identified 15 loci/expressed mRNAs validate as corresponding with outcome in this independent cohort.

4. The authors looked for vulnerabilities in lung squamous by overlapping their profiling results with Project DRIVE. Some attempt at validating AKR1C3, PSAT1 and TP63 in lung squamous cell lines is warranted to confirm they are indeed vulnerabilities. Otherwise this effort is correlative and speculative in nature.

Reviewer #1 (Remarks to the Author):

Stewart et al present a comprehensive proteogenomic study of squamous cell lung cancer (SCC) including DNA copy number, somatic mutations, RNAseq and proteomics data of 108 cases. They describe three proteomic subtypes which they call “inflamed”, “redox” and “wnt/stromal” that partially overlap with transcript-based subtypes. While signatures of inflammatory cells were observed in the inflamed subtype the redox subtype harbored mutations in the genes NFE2L2/KEAP1 and copy number increases in TP63, SOX2 and PIK3CA. No correlations of proteomic subtypes with survival were found, yet by histological examination tertiary lymph node features were found that correlate with good prognosis. However, no signals of these tertiary lymph nodes were explored/found in the proteomics or RNAseq data of matching tumor specimens. This aspect should be either further analyzed or if no correlation can be found further discussed.

This was a good suggestion – the re-examination of tissue histology was provoked by our proteomics data findings but we did not return to see if we could find specific markers associated with TLN. To now address this, we stratified patients by their TLN status (no TLN or TLN score 0 vs. TLN high or TLN score ≥ 2) and used a Wilcoxon rank-sum test adjusted p-value to test for differences of B-cell markers in the RNAseq and proteomics datasets. We have added these results to the text describing TLN results on page 12:

‘TLN exhibited strong staining of CD20 (n = 103), which is consistent with high concentration of B-cells (Figure 4E, 4F). There was significantly higher CD20 staining in TLN ≥ 2 compared to TLN = 0 (P = 0.001). RNAseq showed significantly higher expression of B-cell markers BLK (1.31 log₂ ratio, P = 1.45E-03), CD79A (1.55 log₂ ratio, P = 1.37E-03), and CD79B (1.30 log₂ ratio, P = 2.91E-03) in the TLN ≥ 2 group, and proteomics showed significantly higher expression of B-cell marker IGLL1 (0.63 log₂ ratio, P = 0.013) in the TLN ≥ 2 group.’

In general, proteomics data can complement genomics data by increasing the specificity for calling tumor associated genes/pathways that overlap between both data types or by highlighting additional features that are not visible in the genomics data. The protein to RNA comparisons could be further explored, such as by a subtype-focused analysis of how SCC-related transcription factor target genes of the transcription factors SOX2, p63 and NFE2L2 are regulated on RNA and protein level. SCC subtypes have been previously associated with target genes for SOX2 and p63 and these disease-related gene activities may be highlighted with better specificity using both RNA and protein data. Since one of the candidate disease genes, KEAP1, is part of an E3 ligase, it would be interesting to test if there are any post-translational regulatory events. This could be addressed by a protein-protein correlation analysis of KEAP1 to other proteins in KEAP1 wt tumors.

We appreciate this suggestion. To address this, we identified target gene lists of SOX2, TP63, and NFE2L2 (BENPORATH_SOX2_TARGETS, PEREZ_TP63_TARGETS, and NRF2_01, respectively) from the Molecular Signatures Database (MSigDB; <https://www.ncbi.nlm.nih.gov/pubmed/16199517>). We then examined the transcript-protein correlations in the Redox and Inflamed subtypes for these gene lists. We have added the following to page 14:

‘Given the importance of SOX2, TP63, and NFE2L2 transcriptional gene programs in the biology of SCC, we next examined mRNA transcript-protein correlations using MSigDB signatures BENPORATH_SOX2_TARGETS, PEREZ_TP63_TARGETS, and NRF2_01, respectively^{21,22} (Table 2). All three average TF target transcript-protein correlations were positive (SOX2: 0.37, TP63: 0.34 and NFE2L2: 0.43), suggesting changes in TF target mRNA transcripts were similar to the overall transcript-protein correlations across the entire cohort ($\rho = 0.38$). We observed a significant difference between the SOX2 TF target correlations in Inflamed compared to Redox while no differences were observed in TP63 TF or NFE2L2 TF target correlations. Although this analysis can inform the degree of the relationship between transcript and protein, it does not quantify the magnitude of expression changes. These results suggest that the SOX2 transcriptional program may be translated more consistently in the Redox subtype.’

We also thank the reviewer for pointing out an interesting question about KEAP1 wt regulation. We performed a protein-protein correlation analysis for the subset of KEAP1 mutant patients and again for the subset of KEAP1 wild type patients. We attempted to select for proteins in the KEAP1 mutant subset that had a significant correlation ($P_{\text{adj}} \leq 0.25$), but did not identify any (all $P_{\text{adj}} > 0.25$). As an alternative, we looked at proteins that had a large difference in correlation (Spearman's correlation > 0.5) between mutant and wild type patients. This has been incorporated on page 13:

'KEAP1 is an E3 ligase that has multiple substrates including NFE2L2. We hypothesized tumors with KEAP1 mutations would be enriched for NFE2L2-independent co-expression. To test the hypothesis, we performed a protein-protein correlation analysis with KEAP1 protein expression in KEAP1 mutant tumors (15 samples total; 15 Redox, 1 Inflamed) and separately in KEAP1 wild type tumors (93 samples total; 37 Redox, 42 Inflamed, 14 Mixed). No proteins were identified with a significant correlation to KEAP1 expression in the KEAP1 mutant tumors ($P_{\text{adj}} \leq 0.25$). However, we identified 107 proteins (**Table S10**) with a large difference in correlation to KEAP1 expression ($\rho \geq 0.5$) between KEAP1 mutant and wild type tumors, and enrichment analysis identified regulation of mitotic cell cycle phase transition ($P_{\text{adj}} = 3.16\text{E-}04$), which is consistent with KEAP1 function⁴².'

In summary, the inflamed and redox proteomic subtypes are very interesting findings and this study provides a useful resource dataset with a coverage of nearly 5,000 proteins across a sufficiently large cohort of >100 SCC samples. While the introduction and the discussion are very well written, the results section and main figures need to be shortened and focused on the key observations. Figure 2 can be moved to supplements and the concept pathway figures be reduced in size.

We appreciate the reviewer's concerns regarding the large amount of information that this comprehensive analysis of lung squamous tumors has provided. As a result, we have moved Figure 2 to Figure S06 as this does represent a supplemental finding. We have made some attempt to shorten text and move figures, however the large amount of data and methods has limited our ability, as has some additional experiments suggested by both reviewers. We can revisit this as needed with editorial guidance.

The row numbers were cut for each pdf page, therefore no row numbers are addressed in this report.

Major points of concern:

Figure 1:

- Put NFE2L2 and KEAP1 next to each other to make it easier to see mutually exclusive mutation events

We have moved KEAP1 mutations adjacent to NFE2L2 mutations in Figure 1.

- How were proteins and pathways selected? Did the authors use any FDR control for calling enriched pathways? Table S14 shows for these hits FDR controlled pvals, but no threshold are indicated here

We apologize for not providing sufficient detail for this result. We identified five groups of proteins from the heatmap clustering. We took the proteins from each group and searched against GO: Biological Processes to yield a list of pathways. The topmost enriched pathway ($P_{\text{adj}} < 0.05$) was used to label the protein clustering in the heatmap. We have adjusted the text in Figure 1 to reflect this, and we have updated the caption to describe this. Table S14 contains enrichment results for differentially expressed proteins between the proteomic subtypes and is not related to the results in Figure 1.

- How do RNAs and proteins correlate for these pathways?

We have added the correlations for these pathways to the caption of Figure 1. The Figure 1 caption now reads:

'Figure 1 – Identification of 3 proteomic subtypes in SCC. 108 patient tumors are displayed as columns, and the 1,000 most variable proteins by absolute median deviation are displayed as rows.

The patient tumors were organized by consensus clustering into five clusters corresponding to three biological subtypes (Inflamed, Redox, and Mixed). There is partial concordance with the Wilkerson *et al.* mRNA-based classifiers of these same samples, but the primitive group is not recapitulated. Mutation status and copy alterations of commonly mutated SCC genes/loci are shown directly above the heatmap. We identified five groups of proteins from the heatmap clustering. We took the proteins from each group and searched against GO: Biological Processes to yield a list of pathways. The topmost enriched pathway with $P_{adj} \leq 0.05$ was used to label the protein clustering in the heatmap. The mean transcript-protein correlations for these pathways using matched RNAseq expression were: 0.46 for neutrophil degranulation, 0.56 for extracellular matrix organization, 0.35 for platelet degranulation, 0.64 for glutathione metabolic process, and 0.60 for bicarbonate transport.'

Page 6: The mass spectrometry MS/MS-based protein FDR at 4.75% is rather high. Is there a reason why no conventional FDR<1% was used? The subset of 4,880 proteins used for proteogenomic analysis has the same FDR?

The procedure used to identify proteins was previously applied to study of colorectal cancer as part of the NCI CPTAC program: Zhang *et al.* Nature (2014) 513: 362-7 (<https://www.ncbi.nlm.nih.gov/pubmed/25043054>) and Slebos *et al.* Scientific Data (2015) 2:150022 (<https://www.ncbi.nlm.nih.gov/pubmed/26110064>). Based on this previous experience, we have applied an initial stringent peptide spectral matching or PSM FDR of 0.1% (as compared with many publications that use 1% for PSM FDR) and we required a minimum of 2 distinct peptides identified for each protein. These criteria gave a protein inventory of 8,300 with a protein level FDR of 4.75%. To increase the number of peptide matches used for TMT quantitation, we limited the second search space to just those 8,300 proteins and expanded peptide matches to include those with a 1% PSM FDR or less. From this protein inventory, we filtered the data to include only proteins that were expressed in at least 10% of the tumor samples, for a total of 4,880 proteins. The protein level FDR for this inventory is 1.3%, which we believe is sufficiently stringent to allow deep proteomic analyses, while minimizing the risk of false negative findings. To avoid confusing readers, we have revised the text in page 6:

'Similar to Zhang *et al.* and Slebos *et al.*, we employed a highly stringent, two-step filtering process and identified 8,300 protein groups (average of 6,570 protein groups per sample; see Methods for details)^{11,15}. Functional analysis and downstream associations with DNA and mRNA alterations were restricted to the 4,880 protein groups that were observed in > 90% of the samples, resulting in a final protein false discovery rate of 1.3%.'

Page 8: "Finally, because of the relevance to immune checkpoint therapy, we also examined expression of PDCD1 (PD-1) and CD274 (PD-L1). This was performed using RNAseq data, as these were not reliably quantified in the proteomics experiments."

Explain why these proteins were not reliably quantified and if those could have been excluded by using a more stringent protein FDR of less than 0.01 for protein identification in first place

PD-1 and PD-L1 are not consistently expressed in lung squamous cell carcinomas; as an example, PD-L1 is detected in 34% of the cohort reported here (<https://www.ncbi.nlm.nih.gov/pubmed/27467949>). Proteomic approaches have been developed to detect these proteins, but the strategy relies on peptide fractionation and targeted mass spectrometry (<https://www.ncbi.nlm.nih.gov/pubmed/28546465>). Therefore, it is unlikely that these proteins will be detected in chemical labeling experiments, which favor detection and quantification of peptides that are observed in all samples as well as those that are expressed at higher levels in the proteome. We are working on a similar strategy for quantification of immune checkpoint proteins, but we are not able to add that dataset to this manuscript.

Figure 4: The finding that tertiary lymph nodes are more frequent in the inflammatory proteomic subtype is very interesting and that TLN correlate with good prognosis. Do the authors know if any of their samples for proteomic analysis contained TLN with high B cell counts? Can any of these TLN signals be identified in RNAseq or proteomics data from corresponding tumor specimens?

The H&E images come from the same patients used in the proteogenomic analysis. Based on our CD20 results and the work of others, TLN consist primarily of B-cells and other immune cells. We are able to detect B-cell markers by RNAseq and proteomics when stratifying by TLN status (please see the above response).

Page 11: “Top elevated proteins of the combined Redox subtype included 5 AKR family members (Table S28). AKRs are indicative of NFE2L2 (NRF2) activity and are overexpressed in both SCC and lung ADC”

Indicate if top elevated proteins in redox subtype are also elevated on transcript level. Since KEAP1 is part of an E3 ligase complex it is interesting to test if redox subtype-based protein features are regulated post-translationally or driven by transcription factor-based gene expression programs.

Please see above for our response to your KEAP1 comment. We have revised the text on page 12 to read:

‘Top elevated proteins of the combined Redox subtype included 5 AKR family members (Table S28), and these proteins were similarly differentially elevated at the transcript level (Table S23).’

Page 12: “We started with 303 well-correlated genes across the entire cohort (correlation of CN to Protein > 0.5 and correlation of RNA to protein > 0.5; Figure 5D; Table S11) and examined these for intersections with 94 unique proteins significantly elevated in the Redox subtype. Next, we mined redundant siRNA activity data from Project DRIVE and identified 1,101 gene knockdowns that significantly impacted at least one SCC cell line (RSA score < -3; Table S21)”

Instead of using arbitrary cut offs across thousands of proteins the authors should use here a statistical approach with significance calling for the correlations. For siRNA screens cut-off values have been described frequently in the field. For the positively scoring cell lines the authors should describe if those cell lines resemble the protein or RNA expression features of redox SCC tumors.

We agree with the reviewer that significance calling can improve the confidence in the findings, therefore we have added an adjusted p-value criterion of ≤ 0.25 , an FDR threshold used by TCGA. As we anticipated, this resulted in 13 fewer genes in the “high correlation” group. We have updated the Venn diagram from Figure 5E to reflect this filtering, and we have updated the text to include the new number and the adjusted p-value threshold. We used the same threshold of RSA < -3 as Project DRIVE authors for identifying vulnerabilities, and we have added text describing this to page 14. We have added coloring to the Project DRIVE plot (Figure 5F and 5G) to denote cell lines with known NFE2L2/KEAP1 alterations.

Minor points of concern:

**Table S01, sheet “Data”: include RNA- and protein-based subtype information
Do proteomic subtypes correlate with gender, smoking status or drug treatment?**

Proteomic subtypes did not correlate with gender or smoking status. This information has been provided in Table S20. Additionally, proteomic subtypes did not correspond to drug treatment because all samples used in the final analysis were treatment naïve (see below response). We have added text to clarify these points to the results on page 7.

‘The subtypes were not associated with clinical variables such as stage, gender, or smoking status (Table S20).’

Page 6: Indicate if SCC tumor collection included samples that were treatment naïve or collected after drug treatment. Also indicate if SCCs studied here were primary tumors or if some of those could be metastasis of head and neck squamous cell carcinomas.

All patient samples used in the final analysis and presented in the manuscript were treatment naïve. Figure S01 contains detail about the sample eligibility. We looked at clinical data and reviewed physician notes to determine that these samples were primary lung cancers; any questionable cases were excluded, such as if a

pre-existing head and neck cancer confused the clinical picture. We have added this text to the results on page 6 to clarify to readers.

‘Patient clinical data and physician notes were reviewed to ensure these samples constituted treatment-naïve, primary lung cancer.’

Figure S02: This CNA heatmap would be more informative if the information was not aggregated across all patients, but if all patients were shown as columns and the chromosome regions as rows. Organize the patients by proteomic subtypes.

We agree with the reviewer that less aggregation would be more informative for the CNA. Since most regions have very few alterations (less than 10% of patients), we display the data organized by proteomic subtype and have revised the figure accordingly, and we have updated Figure S02 accordingly.

Page 6: Description of content of supporting data tables S03-S08 and S22-24 is missing.

We have added the description of these supplemental tables to the beginning of the results on page 6.

‘Supporting data can be found in tables **S03** (targeted exome sequencing summary), **S04** (copy number variation counts), **S05** (summarized copy number status), **S06** (proteomics results), **S07** (proteomics metadata), **S22** (gene-level transformation of copy number data), **S23** (RNAseq results), and **S24** (RNAseq metadata).’

Page 8: “Inflamed A, Inflamed B expressed significantly higher levels of 11 MHC Class II proteins and 9 cathepsins”

Describe statistical test used and how significance was assessed.

All differences in protein expression were determined by a non-parametric Wilcoxon rank-sum test. We considered a protein to be differentially expressed if it had at least \log_2 fold-change greater than or equal to 1.5 and an adjusted p-value less than or equal to 0.05. We have reiterated this information from the methods section in the results section on page 8 for clarity.

‘The first cluster (**Figure 1**), Inflamed A, contained 23 tumors (21% of tissues) and presented significantly higher expression of neutrophil-associated proteins compared to the rest of the cohort by a Wilcoxon rank-sum test with ± 1.5 fold-change and $P_{\text{adj}} \leq 0.05$ (MPO, DEFA1, DEFA3, LTF, ELANE, MMP9, and RETN; **Table S28**)^{17,18}. We used a ± 1.5 fold-change threshold since isobaric labeling can lead to underestimation of fold changes¹⁹.’

Table S07: TMT metadata table should also contain information about the RNA and proteomic subtypes to help understand if there any subtypes overlapping with TMT6-plexes, which can be a source of batch effects

We took extensive care to reduce sources of technical variation and batch effects through careful design of experiments and randomization of samples. The Figure S03B principal component plot shows there are no obvious batch effects in the proteomics data. Proteomic subtypes were included in old Table S10, but we have now consolidated this information into Table S07 (TMT metadata).

Table S11: Add p-values and FDR corrected values for correlations, that quality of correlation can be better assessed

We have added p-values and FDR corrected values to Table S11.

Table S13: How were those top proteins per subtype selected? Was any validation attempted by immunohistochemistry?

The top proteins were selected by sorting the fold changes of the differentially expressed proteins. Although not the top differentially expressed proteins, we have examined immune involvement using immunohistochemistry of CD8 (T-cells), CD20 (B-cells), and CD33 (myeloid lineage). We have added a new Figure 3 and have added text describing these results to pages 9-10:

'We hypothesized that the neutrophil signatures generated by our proteomics, pathway, and CIBERSORT analyses represents a broader myeloid type cell enrichment, including not only mature neutrophils but also myeloid derived suppressor cells, monocytes, and macrophages. This may have important ramifications for immunotherapies since other studies have observed neutrophils dominate the immune landscape in lung cancer and have lymphocyte-suppressing capabilities^{34,35}. To address this, we performed CD33 immunohistochemistry, scoring both intratumoral and stromal CD33, using a dual color CD33 plus CD8 assay applied to 22 Inflamed A tumors and 19 Inflamed B tumors. We observed good correlation between CD8 IHC staining and CD8 measured by mass spectrometry ($\rho = 0.49$, $P = 3.60E-03$). We observed more CD33 positive cells in the tumor stroma (**Figure 3, Table 3**) in the Inflamed A subtype while there were comparable amounts of intratumoral CD33 positive cells between the two subtypes. Overall, these results are consistent with Kargl *et al.* who found large areas of NSCLC infiltrated by CD45 + cells with nearly 50% of these CD45+ cells being of myeloid lineage³⁵. Together, these observations suggest that agents that target these large groups of myeloid cells, possibly using agents such as gemtuzumab ozogamicin to target CD33+ cells, could potentially augment immune checkpoint therapy by eliminating immune suppressive cells from the tumor microenvironment.'

Table S26, column H: the trypsin amount should read micro-gram instead of milligram

The typo has been corrected.

Figure 2: RNA-protein correlation similar to previous proteogenomic studies. Move to supplements.

We have moved this to new Figure S06.

Page 7: Does the mixed/stromal subtype show less tumor cellularity?

We have added tumor cellularity data to Table S07. We added the following to page 16:

'There were no differences in tumor cellularity among the subtypes (Wilcoxon rank-sum P-values > 0.05; **Table S07**).'

Page 7: For RNAseq to protein comparison use the term transcript-protein pairs instead of gene-protein pairs

We have changed all mentions of gene-protein to transcript-protein.

Page 8: Describe in text what supporting data tables S09-S11 and S12-S15 contain

We have added description of these tables to page 8.

'Supporting data found in tables **S12** (protein differential expression), **S13** (top proteins by proteomic subtype), **S14** (GeneGO pathway enrichment), and **S15** (MSigDB pathway enrichment).'

Page 10: "We identified 15 such proteins ($q \leq 0.3$; Figure S07), suggesting new prognostic genes and proteins associated with disease recurrence. We identified high expression of the tyrosine phosphatase PTPN12 associated with poor survival which functions by counteracting oxidative stress and promoting cell survival and metastasis in breast cancer 33."

Please explain if q-val or FDR was here selected to be <30%? If that is the case these findings should

not be highlighted in main text. PTPN12 has been also described as a tumor suppressor counteracting receptor tyrosine kinases for example in triple negative breast cancer. Describe why PTPN12 acts in this context as an oncogene?

We feel these findings are important to report for completeness, however we agree with the reviewer that the significance of these results was not adequately stated. We have dropped the comment regarding prognostic genes. We agree that PTPN12 has been described as a tumor suppressor. Interestingly, in (<https://www.ncbi.nlm.nih.gov/pubmed/23435421>) the authors report that patients whose tumors displayed high levels of PTPN12 transcripts had significantly poorer prognosis and therefore may be oncogenic. We have rephrased the wording on page 11 as follows:

‘We next tested for associations with our three proteomic subtypes with the extensive clinical, pathology, and outcome data collected for our cohort, but we identified no significant differences in overall or recurrence-free survival in our cohort (**Figure 4A**). This was surprising given the immune phenotype observed in the Inflamed tumors, which we assumed would be associated with better outcomes. We next tested top mutated genes (e.g. *TP53*, *MLL2*, *NFE2L2*) but did not find any association with outcomes. We were also unable to identify associations with outcome in the DNA copy number, mRNA abundance, and protein abundance datasets. These negative findings are in line with the general lack of agreement of prognostic signatures in SCC^{3,37}.

We attempted to identify coordinated associations of patient outcome with DNA copy number, mRNA abundance, and protein abundance through a combined meta-analysis of these datasets. A meta-analytic random-effects model was fitted using empirical Bayes as the heterogeneity estimator using gene-level CNV, RNAseq expression, and protein expression from 3,484 genes present in all three datasets. However, we had to relax FDR to 30% (Storey’s q-value ≤ 0.3) in order to identify 15 proteins (**Figure S08**) associated with outcome. This meta-analysis identified high expression of the tyrosine phosphatase PTPN12 associated with poor survival. PTPN12 has been described as a tumor suppressor that counteracts receptor tyrosine kinase signaling and oxidative stress, but it has also been shown to support tumorigenesis and was associated with poor prognosis in breast cancer³⁸. We performed two exploratory analyses to test if these genes impacted survival in the TCGA SCC cohort. The first utilized RNAseq expression and stratified by median expression the second stratified copy number alteration or mutation status, but both failed to identify any significant associations³⁹.

Page 11: “We took samples with either a NFE2L2 or KEAP1 mutation (41 samples total; 32 Redox, 6 Inflamed, 3 Mixed), and when compared to wild type tumors, we found that mutated tumors were enriched for oxidative stress and metabolism pathways (Figure 5C).”

This analysis strongly overlaps with the redox subtype. Do the associated proteins have antioxidant response elements in their respective genes and constitute NFE2L2 target genes?

The antioxidant response element (ARE) possesses inherent flexibility, and ARE sequences are dependent on binding partner. There is no single method that can predict/identify ARE sequences at the genome level, and they differ between organisms (e.g. you cannot necessarily apply mouse gene lists to human). We have clarified that stratifying by NFE2L2/KEAP1 status is just another way to recapitulate the Redox subtype on page 13:

‘To clearly show the mutant NFE2L2/KEAP1 proteome was analogous to Redox, we next took tumors with either a NFE2L2 or KEAP1 mutation (41 samples total; 32 Redox, 6 Inflamed, 3 Mixed) and compared to wild type tumors. This stratification enriched for Redox tumors, and we observed downstream effects of NFE2L2 signaling such as enrichment of oxidative stress and glutathione pathways (**Figure 5C**).’

No attempt was described if any tumor-specific single amino acid variants were detected on protein level.

We did not report the presence or absence of single amino acid variants (SAAVs). Previous reports have examined personalized database searches to detect protein evidence for mutation status in the form of SAAVs.

Large scale proteomics datasets have typically yielded less than 100 SAAVs, some of which will be related to cancer drivers and others will be passenger mutations that are not relevant. These previous publications have stopped at the presentation of the catalog of mutations and have not investigated their functional significance. Because we have pursued a general classification of lung squamous cell tumors using 6-plex TMT, we expect to have even less possibility of detecting SAAVs in these experiments than in the label free and 4-plex iTRAQ datasets generated by the CPTAC due to dilution by the use of 2 pooled samples mixed with each set of 4 individual tumors. We do plan to pursue this avenue in other datasets, because it will be important in the future as we develop more molecular detail for these signatures delineating types of lung squamous cell carcinomas or search for neo-antigens.

Reviewer #2 (Remarks to the Author):

In the manuscript by Stewart et al, entitled “Proteogenomic Landscape of Squamous Cell Lung Cancer”, the authors detail an impressive single institution effort to characterize lung squamous tumors based on several -omic platforms. The generated data is integrated and then several approaches are taken to better understand the various biologic phenomena of different proteomic subsets. The authors are to be commended for their effort given its scope and the resource it will create for the field. Although this work will be interest to the field, some aspects are often descriptive and, in some cases, not enough has been done to validate the findings.

We appreciate the commendation on our work from this reviewer and appreciate the suggestions for improving the paper which we describe in detail below.

General Comments:

1. In the Inflamed A+B subtypes, the authors find enrichment for markers of neutrophils and their degranulation. The authors are encouraged to perform IHC and score at least a representative subset of tumors from each cluster and assess for this.

Based on the reviewer suggestion, we performed CD33 immunohistochemistry using a dual color CD33 plus CD8 assay applied to 22 Inflamed A tumors and 19 Inflamed B tumors. We have added the details to pages 9-10:

‘We hypothesized that the neutrophil signatures generated by our proteomics, pathway, and CIBERSORT analyses represents a broader myeloid type cell enrichment, including not only mature neutrophils but also myeloid derived suppressor cells, monocytes, and macrophages. This may have important ramifications for immunotherapies since other studies have observed neutrophils dominate the immune landscape in lung cancer and have lymphocyte-suppressing capabilities^{34,35}. To address this, we performed CD33 immunohistochemistry, scoring both intratumoral and stromal CD33, using a dual color CD33 plus CD8 assay applied to 22 Inflamed A tumors and 19 Inflamed B tumors. We observed good correlation between CD8 IHC staining and CD8 measured by mass spectrometry ($\rho = 0.49$, $P = 3.60E-03$). We observed more CD33 positive cells in the tumor stroma (**Figure 3, Table 3**) in the Inflamed A subtype while there were comparable amounts of intratumoral CD33 positive cells between the two subtypes. Overall, these results are consistent with Kargl *et al.* who found large areas of NSCLC infiltrated by CD45 + cells with nearly 50% of these CD45+ cells being of myeloid lineage³⁵. Together, these observations suggest that agents that target these large groups of myeloid cells, possibly using agents such as gemtuzumab ozogamicin to target CD33+ cells, could potentially augment immune checkpoint therapy by eliminating immune suppressive cells from the tumor microenvironment.’

2. Several of the observations are consistent with prior work already in publication. While this is important since it comes from an independent cohort, incorporates proteomics (which none others have) and adds rigor to the field, the authors are encouraged to cite and, in some cases (per below, point #c), expound on such instances.

a. Higher PDCD1 (and several other immune checkpoints) but not CD274, have been shown by RNAseq to be higher in the Secretory subtype of lung squamous (Faruki et al, Journal of Thoracic Oncology, 2017)

b. Enrichment of the Inflamed A+B subtypes, which largely consist of Secretory and Basal subtypes, have recently been shown to be highly enriched for and driven by monocytes (Porrello et al, Nature Communications, 2018)

c. In Korean lung squamous cancers (Seo et al, Cancer Immunology Research, 2018), they also characterize an inflammatory subtype (Subtype B in their case), that similarly enriches for Secretory and Basal subtypes. This study looked at somatic copy number variations and found these are inversely correlated with Subtype

Thank you for pointing out these omissions. We have added these citations and refer to them in the discussion of our results on page 18:

'PD-1 and other immune checkpoints, but not PD-L1, have been shown to have elevated gene expression in the secretory group⁷⁴. We observed significantly higher PD-1 RNA expression in the combined Inflamed subtype, which is potentially indicative of exhausted T-cells⁷⁵, but we did not detect PD-1 or PD-L1 in our proteomics experiments. We identified significantly higher monocytes in the Inflamed subtype (**Figure 2C**), and this is consistent with a recent observation that the secretory and basal groups are highly enriched for and driven by monocytes⁷⁶.

B. Along these lines, the authors are encouraged to assess whether SCNVs have similar inverse correlations with their Inflamed subtypes.

In our results on page 10, we describe how the Inflamed subtype was enriched for a single amplification on 14q3 and how Inflamed had significantly less amplification of the 3q2 locus. We have added the citation suggested by the reviewer because we believe it enhances our results and adds context to this finding. We have revised the old Figure 2 (now Figure S06) to display the breakdown of copy number gains/losses by subtype, which also helps make this point more clear.

3. Please elaborate more on how the meta-analysis was performed using CNA, mRNA and protein data to relate with patient outcome. Were these the top 15 survival hits for all available loci/mRNA/proteins?

We have revised our explanation of the meta-analysis to include more detail. Yes, these top 15 survival hits were available in gene-level CNV data, RNAseq, and proteomics. This section on page 11 now reads:

'We next tested for associations with our three proteomic subtypes with the extensive clinical, pathology, and outcome data collected for our cohort, but we identified no significant differences in overall or recurrence-free survival in our cohort (**Figure 4A**). This was surprising given the immune phenotype observed in the Inflamed tumors, which we assumed would be associated with better outcomes. We next tested top mutated genes (e.g. *TP53*, *MLL2*, *NFE2L2*) but did not find any association with outcomes. We were also unable to identify associations with outcome in the DNA copy number, mRNA abundance, and protein abundance datasets. These negative findings are in line with the general lack of agreement of prognostic signatures in SCC^{3,37}.

We attempted to identify coordinated associations of patient outcome with DNA copy number, mRNA abundance, and protein abundance through a combined meta-analysis of these datasets. A meta-analytic random-effects model was fitted using empirical Bayes as the heterogeneity estimator using gene-level CNV, RNAseq expression, and protein expression from 3,484 genes present in all three datasets. However, we had to relax FDR to 30% (Storey's q-value ≤ 0.3) in order to identify 15 proteins (**Figure S08**) associated with outcome. This meta-analysis identified high expression of the tyrosine phosphatase PTPN12 associated with poor survival. PTPN12 has been described as a tumor suppressor that counteracts receptor tyrosine kinase signaling and oxidative stress, but it has also been shown to support tumorigenesis and was associated with poor prognosis in breast cancer³⁸.

Along these lines, since CNA and RNAseq data and clinical outcome are publicly available with TCGA, the authors are encouraged to assess whether any of the identified 15 loci/expressed mRNAs validate as corresponding with outcome in this independent cohort.

We performed a survival analysis using the RNAseq-based gene expression from TCGA SCC tumors, but we did not find any of the genes to be significant. We performed another survival analysis using combined copy number alteration and mutation data, but again we did not find any significant genes. These negative findings have been added to page 11:

'We performed two exploratory analyses to test if these genes impacted survival in the TCGA SCC cohort. The first utilized RNAseq expression and stratified by median expression the second stratified copy number alteration or mutation status, but both failed to identify any significant associations³⁹.

4. The authors looked for vulnerabilities in lung squamous by overlapping their profiling results with Project DRIVE. Some attempt at validating AKR1C3, PSAT1 and TP63 in lung squamous cell lines is warranted to confirm they are indeed vulnerabilities. Otherwise this effort is correlative and speculative in nature.

We have performed additional validation experiments as suggested by analyzing the Pooled In-vitro CRISPR Knockout Library Essentiality Screens (PICKLES) database for squamous cell lines. These confirmatory results have been added to page 15:

‘We performed an additional validation experiment by querying Pooled In-vitro CRISPR Knockout Library Essentiality Screens (PICKLES) for the same genes and cell lines used in our Project DRIVE Analysis (5 cell lines and 5 genes except SERPINB5 were available). We used the same Bayes factor (BF) of > 3 for a threshold of significance as the PICKLES authors⁴⁴. The three genes consistently deleterious in both datasets, PSAT1, TP63, and TFRC, are biologically relevant to metabolic signaling within the Redox subtype and pose potential vulnerabilities in SCC. PSAT1 is regulated by NFE2L2 and catalyzes serine biosynthesis, which is important for the growth of NFE2L2/KEAP1 mutant lung cancer cell lines⁴⁵. TP63 represents the Δ Np63 (or DNp63) isoform, which is the dominant isoform in SCC^{5,46}. Δ Np63 has lost its N-terminus and has altered exon use to produce an oncogenic protein⁴⁷. TP63 amplification was shown to upregulate glutathione metabolism, which is enriched in Redox (**Figure 5A**), and promote tumorigenesis⁴⁸. Deletion of the Δ Np63 or Δ Np73 isoforms in TP53-deficient tumors led to metabolic reprogramming and tumor regression⁴⁹. NFE2L2 regulates iron homeostasis, and in turn TFRC is involved in cellular uptake of iron and is overexpressed in many cancers^{50,51}. Taken together, these results suggest that Redox tumors harbor metabolic vulnerabilities that could be therapeutically exploited.’

Reviewers' comments:

Reviewer #1 (Remarks to the Author):

The authors have properly addressed all points that needed improvement

Reviewer #2 (Remarks to the Author):

The authors have satisfactorily addressed all of my comments, and concerns, and I have no further issues I believe need to be addressed.

Reviewer #3 (Remarks to the Author):

Stewart et al. described the first large-scale proteogenomic characterization of the LUSC cohort of 108 patients that will be an invaluable resource for the research community. While I acknowledge the need to publish such resources quickly, statistical analyses of the data and presentation of the key results need to be substantially improved to clearly communicate the impact of this work.

Comments:

1. In addition to meta-analysis, none of the statistical tests performed were described in the Methods section and such details are required. For example, what are the parameters of consensus clustering? Perhaps Figure S4 provides some insights but how exactly were the 5 clusters determined? What kind of survival analyses were performed? This comment applies to all the following analysis. It would be best to provide such clarification and GitHub directory to the scripts used to ensure reproducibility of the authors' findings. Most statistical analyses the authors performed seem to be standard but until such details are provided it is hard to evaluate the bioinformatics integrity of the study.
2. The authors are to be applauded on presenting the negative result of proteomic subtype not correlating with survival (as scientists typically refrain from doing). However, proteomics subtype is associated with TLN; and TLN is associated with survival. More investigation into this is required; for example, what happens when instead of (ex. Cox model with both proteomics subtype and TLN as predictor variables).
3. Immune-subtype tumors are less likely to have mutations in KEAP1/APC or other driver genes, but how about overall mutation rates? In many cancer cohorts, tumor mutational burden was associated with higher immune infiltrates and better outcome after immunotherapy. The header statement "fewer gene mutations/CNVs" needs to be backed by analyses associating to the whole genome/exome mutation/CNV counts. If that's still the case, potential discrepancy or consistency with known findings in lung cancer should be discussed.
4. Cibersort has been known to produce unstable cell type results. Have the authors made sure their deconvoluted cell type fractions are stable by either experimenting with different cibersort runs, use other tools such as xCell, or in the least show the validation of cell type biomarker through IHC/RNASeq in a more parallel manner to make sure the key conclusion holds?
5. The authors described the general landscape of CNV/transcript/protein correlation. It would be very helpful for the readers if they can zoom in to the multi-omics landscape of genes important in squamous cell lung cancer like in Mertins et al (PMID: 27251275) Figure 1C.
6. Why did the authors look for only highly correlated genes for inhibition targets? Shouldn't any overexpressed proteins present as likely drug targets? Also, it seems like inhibition targets should be identified for all subtypes of cancers and the best targets in each subtype can be further compared.
7. The PCs in Figure 6 do not seem to be particularly helpful especially given the small amount of variance explained.
8. The writing needs to be edited for clarity (even the abstract!) and multiple grammar errors

should be corrected.

Signature: Kuan-lin Huang

1*. In addition to meta-analysis, none of the statistical tests performed were described in the Methods section and such details are required. For example, what are the parameters of consensus clustering? Perhaps Figure S4 provides some insights but how exactly were the 5 clusters determined? What kind of survival analyses were performed? This comment applies to all the following analysis. It would be best to provide such clarification and GitHub directory to the scripts used to ensure reproducibility of the authors' findings. Most statistical analyses the authors performed seem to be standard but until such details are provided it is hard to evaluate the bioinformatics integrity of the study.

We provided the parameters used for consensus clustering in the methods under the old "Integrative molecular data analysis" subsection in the Bioinformatics section of the methods. We have added the following text to the methods on page 11 to help clarify how $k = 5$ consensus cluster groups were chosen:

'Consensus clustering output provides cluster assignments and stability evidence for the identified k clusters. The process is repeated for multiple values of k . The purpose of the CDF plot (Figure S04A) is to find the smallest k at which the change of the area under the CDF curve (Figure S04B) is small, suggesting that there is not much benefit to further divisions. Here, we chose $k = 5$ since it was the first point which corresponded to less than 10% change in area, and larger k would have only given small additional increases in area.'

We provided details for the statistical tests performed under "Integrative molecular data analysis" subsection in the Bioinformatics section of the methods. Based on concerns regarding the details, we have reorganized this section into smaller sections to make it easy to find what statistical tests were utilized. The key finding of proteomic subtypes in SCC relied on consensus clustering, which we have now described in more detail (see above). Most differences between the subtypes (e.g. protein expression, CIBERSORT scores) were determined using a non-parametric Wilcoxon rank-sum/Mann-Whitney U tests and is also described in "Integrative molecular data analysis". Table S20 - Association tests and tables.docx has details related to association tests and survival analyses, but we have also added the following text to the methods on page 12 to describe the survival analyses in further detail:

'*Survival analysis.* Cox regression for overall survival (OS) and recurrence free survival (RFS) was fitted to gene-level CNV, RNAseq expression, and protein expression data. Time-to-event for OS was time in years from date of sample collection to date of death and censored at date of last contact, and time-to-event for RFS was time in years from date of sample collection to date of first clinically confirmed recurrence if a patient recurred or died and censored at the date of last contact. The event for OS was death and for RFS was recurrence or death.'

We agree reproducibility is important, which is why we have included extensive supplemental tables (28) with all of the data necessary to reproduce the findings in our manuscript. Based on your comment, we have provided scripts used for the main findings and the survival analyses in a public GitHub repository here: https://github.com/pstew/proteogenomics_scc . We cannot provide R objects with all data at this time because these could contain protected health information (PHI) and would require internal approvals before release.

2. The authors are to be applauded on presenting the negative result of proteomic subtype not correlating with survival (as scientists typically refrain from doing). However, proteomics subtype is associated with TLN; and TLN is associated with survival. More investigation into this is required; for example, what happens when instead of (ex. Cox model with both proteomics subtype and TLN as predictor variables).

We agree that this finding is somewhat complex. Although tumors with TLN were enriched in the inflamed subtype, TLN was observed in all 3 proteomic subtypes at different frequencies. Specifically, the frequency is: 30 (69.7%) Inflamed, 26 (51%) Redox, and 7 (50%) Mixed (added to results section, page 12). Thus, the distribution of TLN across subtypes suggests that this effect is not completely overlapping the Inflamed subtype. As a result, TLN but not subtype is associated with survival.

3. Immune-subtype tumors are less likely to have mutations in KEAP1/APC or other driver genes, but how about overall mutation rates? In many cancer cohorts, tumor mutational burden was associated with higher immune infiltrates and better outcome after immunotherapy. The header statement “fewer gene mutations/CNVs” needs to be backed by analyses associating to the whole genome/exome mutation/CNV counts. If that’s still the case, potential discrepancy or consistency with known findings in lung cancer should be discussed.

The focus of the manuscript was on key genes known to be involved in SCC. We did not perform whole exome sequencing; therefore, we cannot determine tumor mutational burden directly. However, other studies have seen no association between tumor-infiltrating lymphocytes and tumor mutational burden (PMID: 30106638). In the targeted genes sequenced there were more nonsynonymous mutations in the Redox group (Inflamed mean mutations 15.8, Redox mean mutation 18) however we feel this may not adequately represent TMB (PMID: 30106638) and therefore focus on key genes. We do agree with the reviewer in being specific and have modified the header statement of the page 10 section to “Inflamed tumors demonstrate fewer mutations of key genes and fewer alterations of key chromosomal regions”.

We have clarified the CNVs demonstrate fewer alterations of key chromosomal regions. A study by Seo *et al.* (cited in the manuscript, page 11) found a negative correlation between immune score and copy number variation in 101 Korean squamous cell lung cancers, which is consistent with our findings.

4*. Cibersort has been known to produce unstable cell type results. Have the authors made sure their deconvoluted cell type fractions are stable by either experimenting with different cibersort runs, use other tools such as xCell, or in the least show the validation of cell type biomarker through IHC/RNASeq in a more parallel manner to make sure the key conclusion holds?

We have provided immunohistochemical validation of the main CIBERSORT findings (results section, pages 9 - 12). Based on your suggestion, we ran xCell on our RNAseq dataset and found the findings similar to our initial CIBERSORT results: xCell-based memory B-cell and neutrophil cell types were significantly higher in Inflamed compared to the other subtypes ($P = 2.53E-04$ and $P = 2.083E-05$, respectively; new Supplemental Figures S07E and S07F, pictured below). Memory B-cells and neutrophils were highly correlated between xCell and CIBERSORT ($\rho = 0.654$ and $P = 1.58E-14$, $\rho = 0.58$ and $P = 4.43E-11$, respectively). The consistency of memory B-cells and neutrophils estimates are consistent with the findings of the xCell manuscript, which shows good correlation between xCell and CIBERSORT for several cell fractions including memory B-cells and neutrophils – see Figure 2C (PMID: 29141660).

We have added these findings to the results on page 10:

‘To address this, we first confirmed our neutrophil finding with xCell, another method for estimating cell infiltration. Similar to CIBERSORT, neutrophils were significantly higher in Inflamed ($P = 2.083E-05$; Figure S07E) compared to the rest of the cohort.’

and page 12:

‘Therefore, we first confirmed our memory B-cell finding using xCell. Similar to CIBERSORT, memory B-cells were significantly higher in Inflamed ($P = 2.53E-04$; Figure S07F) compared to the rest of the cohort.’

5. The authors described the general landscape of CNV/transcript/protein correlation. It would be very helpful for the readers if they can zoom in to the multi-omics landscape of genes important in squamous cell lung cancer like in Mertins et al (PMID: 27251275) Figure 1C.

Thank you for the suggestion. After consideration, we believe that presenting a more global view of the data is appropriate. Given limited information on genomic drivers in SCC, a targeted figure may not adequately represent the full description of squamous cell lung cancer that the global heatmap provides.

6. Why did the authors look for only highly correlated genes for inhibition targets? Shouldn't any overexpressed proteins present as likely drug targets? Also, it seems like inhibition targets should be identified for all subtypes of cancers and the best targets in each subtype can be further compared.

We present one approach for analyzing and interpreting the data with a focus on identifying opportunities for inhibition specific to the discovered subtypes. This question is an important one to be addressed and it is for these reasons that we have purposefully provided all data to make this a resource for the community. We anticipate many additional discoveries are possible from this data.

7*. The PCs in Figure 6 do not seem to be particularly helpful especially given the small amount of variance explained.

Our rationale for creating the principal component (PC) plot was to use an independent visualization to verify our findings from consensus clustering. We agree that the explained variances were small for PC1 (12.5%) and PC2 (5.1%), but we note the amount of variance explained by PC's 1 and 2 is small

due to the size of the input matrix (4880 proteins by 108 patients) with inclusion of all protein variables in the PCA model. To show that the first two PC dimensions encapsulate the findings of our manuscript and that we are not excluding important information from the other principal components, we took the first 50 principal components of the input matrix (83.26% variance explained) and performed tSNE clustering with the Rtsne R package. The tSNE-based clustering shows the same grouping of patients by subtypes as the PCA visualization for a range of perplexities:

8*. The writing needs to be edited for clarity (even the abstract!) and multiple grammar errors should be corrected.

Two co-authors (SE and JK) have carefully gone through the manuscript and made additional edits for clarity. We would be happy to enlist editorial assistance to help us further revise the wording and grammar in the manuscript.

REVIEWERS' COMMENTS:

Reviewer #3 (Remarks to the Author):

The authors have effectively improved the manuscript and properly addressed my concerns.